# ESI: Epistemic Uncertainty Quantification via Semantic-preserving Intervention for Large Language Models

## Abstract

Uncertainty Quantification (UQ) is a promising approach to improve model reliability, yet quantifying the uncertainty of Large Language Models (LLMs) is non-trivial. In this work, we establish a connection between the uncertainty of LLMs and their invariance under semantic-preserving intervention from a causal perspective. Building on this foundation, we propose a novel grey-box uncertainty quantification method that measures the variation in model outputs before and after the semantic-preserving intervention. Through theoretical justification, we show that our method provides an effective estimate of epistemic uncertainty. Our extensive experiments, conducted across various LLMs and a variety of question-answering (QA) datasets, demonstrate that our method excels not only in terms of effectiveness but also in computational efficiency.

## 1 Introduction

The recent development of Large Language Models (LLMs) has transformed them from an experimental technology to an everyday tool (Naveed et al., 2023; Zhao et al., 2023; Chiarello et al., 2024). Nevertheless, even the most capable LLM still frequently generates untruthful content, commonly known as the hallucination phenomenon (Zhang et al., 2023; Huang et al., 2024), which significantly degrades their reliability and limits applicability.

Uncertainty Quantification (UQ) is considered a promising direction to improve the reliability of LLMs. The estimated uncertainty score can be applied in various domains, such as error detection, active learning, and selective generation (Zhang et al., 2023; Baan et al., 2023). Uncertainty is generally categorized into two types: aleatoric (as known as data) uncertainty and epistemic (as known as model) uncertainty (Hüllermeier & Waegeman, 2021). Aleatoric uncertainty stems from the inherent, irreducible randomness within the data (e.g., multiple plausible answers exist for a given question). Epistemic uncertainty arises from the lack of knowledge of the underlying ground-truth data-generating process, which can be attributed to factors such as insufficient training or distribution shifts between training and test sets. Epistemic uncertainty is often considered a more reliable indicator of model trustworthiness (Xiao & Wang, 2021; Baan et al., 2023).

Although UQ's effectiveness has been demonstrated on classification and regression tasks (Gawlikowski et al., 2023), UQ for free-form generation is not straightforward. The challenge stems from the intrinsic nature of natural languages, such as the intractable output space (Lin et al., 2024) and entanglement of epistemic and aleatoric uncertainty (Baan et al., 2023). Existing literature on UQ in free-form generation predominantly relies on measuring the semantic variation within the output space through sampling (Kuhn et al., 2023; Duan et al., 2024; Chen et al., 2024), as shown in the bottom-left graph in Figure 1. These methods generally require a large sample size to reconstruct the intractably vast output space, and also estimate total uncertainty rather than epistemic uncertainty.

In this paper, we introduce a graphical causal model of the language generation process, grounded in the assumption that human responses are primarily causally determined by the semantics of the context text. This leads to the conclusion that the ground-truth language-generating function should remain stable under semantic-preserving interventions. Building on this causal perspective,

we establish a connection between the uncertainty of LLMs and the strength of the causal pathways that govern their inference: the better a model captures the underlying causal mechanism, the more stable it is under semantic-preserving interventions, and the lower its uncertainty. Based on this premise, we propose a novel method for UQ in LLMs, **E**pistemic uncertainty quantification via **S**emantic-preserving **I**ntervention (**ESI**), which measures output invariance under semantic-preserving interventions applied to prompts. Specifically, we quantify the average shift in the token predictive distribution of the same response before and after semantic-preserving interventions, as illustrated in the bottom-right graph of Figure 1.

We provide a theoretical justification showing that our method serves as an effective estimator of epistemic uncertainty rather than total uncertainty. Unlike prior approaches, our method circumvents the need to reconstruct the vast output space, leading to greater stability and efficiency. Through extensive experiments across four models and five datasets, we demonstrate that ESI consistently outperforms state-of-the-

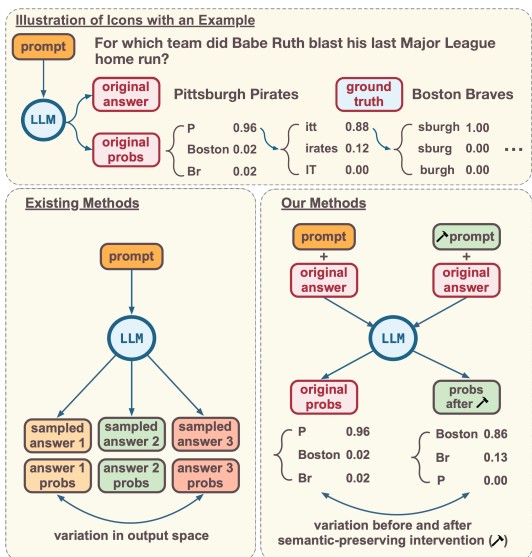

Figure 1: Illustration of our proposed ESI approach and existing uncertainty quantification methods. The term 'probs' refers to the predictive distributions generated by LLMs. The little hammer refers to the semantic-preserving intervention.

art (SOTA) methods, particularly excelling in datasets characterized by stronger causal relationships between inputs and outputs or higher levels of aleatoric uncertainty. Furthermore, our efficiency analysis shows that our approach is more computationally efficient, which not only reduces computation time (by 3-5 times) for the same sample size but also achieves good performance with fewer samples (as few as 2-3 samples).

## 2 UNCERTAINTY QUANTIFICATION VIA SEMANTIC-PRESERVING INTERVENTION

In this section, we demonstrate our motivation from a causal perspective and build a bridge between the uncertainty of LLMs and their invariance under semantic-preserving intervention.

We firstly introduce a graphical causal model to explain the generation process of a text pair $(C, R)$, inspired by the *double triangle of language production* theory proposed by Baan et al. (2023), see Figure 2. The solid arrow $X \rightarrow Y$ suggests that $X$ is the cause of $Y$, while the dashed line indicates that the two variables are correlated but not causally related.

A speaker, on the left in Figure 2, generates the text $C$ based on their intended semantic meaning $S_{\text{intend}}$ and a context variable $U_C$. The context variable $U$ represents all additional factors contributing to the text construction process, such as the lexicon and grammar of the speaker's language, their personal language usage habits, or the bias in their mind (Sun et al., 2024). The responder interprets the

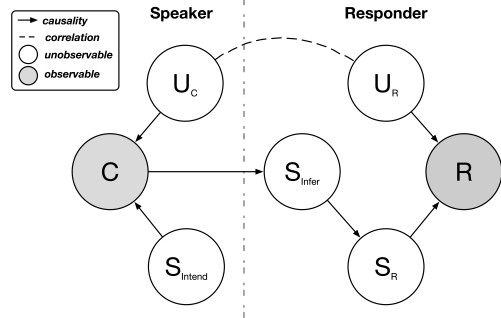

Figure 2: The graphical causal model of the data generating process of a text pair: context $C$ and response $R$.

context text $C$ into the inferred semantics $S_{\text{infer}}$. They then causally deduce an appropriate response semantics $S_R$, and finally generate the response text $R$ with the responder's context variable $U_R$.

We can observe two sources of correlation between $C$ and $R$ from Figure 2, one from the correlation between $U_C$ and $U_R$, and another from the causal path $C \rightarrow S_{\text{infer}} \rightarrow S_R \rightarrow R$. Although $U_C$ and

$U_R$ are correlated, they are not causally related, as they share many common confounders, such as shared language usage (lexicon and grammar). Our proposed graphical causal model implies a basic assumption on the ground-truth language-generating function: The generated response $R$ is causally determined by the semantics of the context text $U_C$.

When training an LLM, we want the model to learn the ground-truth language generation function, i.e., the underlying semantic causal relationships between the context $C$ and the subsequent text $R$ ($C \rightarrow S_{\text{infer}} \rightarrow S_R \rightarrow R$), rather than merely capturing superficial correlations between the context variables $U_C$ and $U_R$, which are mostly known as the spurious correlation (Arjovsky et al., 2019). However, the statistical learning paradigm does not ensure this as it learns correlation instead of causation. (Arjovsky et al., 2019). When a model primarily relies on spurious correlations to infer its answer, the answer is unreliable and vulnerable to out-of-distribution inputs (Geirhos et al., 2020; Arjovsky et al., 2019). Conversely, when the model's inference is based on a causal pathway, its response is likely to be more reliable and confident. It is analogous to humans: individuals tend to exhibit lower uncertainty in their responses when they understand the underlying causal mechanisms, whereas they are more uncertain when based on mere clues or patterns. Therefore, we postulate that **the uncertainty of LLMs can be effectively quantified by evaluating the degree to which the model relies on semantic causal relationships in its inference process.**

But how can this reliance be quantified? Fortunately, based on our causal model in Figure 2, if a prediction is generated through the causal pathway (likewise the ground-truth function), it should be invariant to interventions on the context $C$, so long as these interventions leave $S_{\text{infer}}$ unchanged, i.e., semantic-preserving. In contrast, if a prediction is based on the correlation between $C$ and $R$ through $U_C$ and $U_R$, it is vulnerable and can be easily blocked by intervening on $C$ (Pearl, 2009). In other words, the uncertainty of LLMs can be measured by the invariance of the output under semantic-preserving interventions on the input.

Formally, let $\mathcal{M}$ represent an LLM, and let $\boldsymbol{x}$ be a prompt, Our goal is to find a function $\mathbf{U}$ that quantifies the uncertainty of $\mathcal{M}$ given the prompt $\boldsymbol{x}$, written as $\mathbf{U}(\mathcal{M}, \boldsymbol{x})$. Based on the analysis outlined above, we can estimate $\mathbf{U}(\mathcal{M}, \boldsymbol{x})$ as follows:

$$\mathbf{U}(\mathcal{M}, \boldsymbol{x}) = E_{\tilde{\boldsymbol{x}} \sim f_I}\big[\mathbf{V}(\tilde{\boldsymbol{x}}; \boldsymbol{x}, \mathcal{M})\big] \tag{1}$$

Here, $f_I$ is the semantic-preserving intervention function that maps the original prompt $\boldsymbol{x}$ to a semantically equivalent variant $\tilde{\boldsymbol{x}}$ with some probability. $\mathbf{V}$ is the function measuring the variation in the model's output before and after the semantic-preserving intervention.

## 3 METHODOLOGY

In this section, we present the details of our ESI method for quantifying the uncertainty of LLMs via semantic-preserving intervention.

### 3.1 SEMANTIC-PRESERVING INTERVENTION

We adopt two approaches to implement the semantic preservation intervention: a sentence-level intervention, Paraphrasing (Para) (Jiang et al., 2023; Hou et al., 2024), and a character-level intervention, Skip-One-Char (SOC).

Following prior research, we leverage few-shot in-context learning to prompt LLMs to generate paraphrases. A well-crafted paraphrase can effectively alter the superficial linguistic structures while preserving the underlying semantic meaning. However, this approach requires a powerful paraphrasing model, generally in a large size, capable of generating diverse and semantically consistent paraphrases, which significantly increases the computational cost.

Therefore, we introduce another simple but effective alternative, Skip-One-Char (SOC), which randomly removes one character from the latter portion of randomly selected words in given prompts:

*For whieh team did Babe Ruth blast his last Majer League home run?*

Formally, let $\boldsymbol{x} = \{\boldsymbol{w}_1, \boldsymbol{w}_2, ..., \boldsymbol{w}_N\}$ represent a text prompt of length $N$, where each $\boldsymbol{w}_n$ is a word consisting of $M_n$ characters, denoted as $\boldsymbol{w}_n = \{\boldsymbol{w}_{n,m}\}_{m=1}^{M_n}$. The intervention function $f_I^{\text{soc}}(\boldsymbol{w}_n)$

operates on each word $\boldsymbol{w}_n$ such that, with probability $p$, one random character $\boldsymbol{w}_{n,k}$ is removed, resulting in $f_1^{\text{soc}}(\boldsymbol{w}_n) = \{\boldsymbol{w}_{n,m}\}_{m \neq k}$, where $k \sim \text{Uniform}(M^*, M_n)^1$. Otherwise, the word remains unchanged with probability $1 - p$, i.e., $f_1^{\text{soc}}(\boldsymbol{w}_n) = \boldsymbol{w}_n$. Here, $M^*$ is a hyperparameter that ensures the function only removes the latter characters from words longer than a specified length, protecting informative tokens. The parameter $p$ controls the proportion of words that are intervened.

To assess the semantic-preserving effectiveness of the two intervention methods, we employ two approaches: a Natural Language Inference model (NLI-judge) and an LLM prompted with judging semantic equivalence (LLM-judge). Each intervention is applied five times per prompt across four datasets, and we compute the average semantic preservation scores. Both judges classify over 90% of the intervened prompts as semantic equivalent for both methods, indicating that both intervention methods effectively preserve the semantics. Details and results are provided in Appendix C.1.

## 3.2 Invariance Measurement Function

To measure the variation in the model's output induced by the intervention, we propose quantifying the average token-wise distribution shift of the same response before and after interventions, see Figure 1. Specifically, we first generate the response and its associated token predictive distributions based on the original prompt. Next, we concatenate the same response with the intervened prompt to obtain the token predictive distributions after the intervention with a single forward pass. Finally, we calculate the distribution shift at each token position and average these values.

Formally, following the same annotation in Equation 1, we use $p_{\mathcal{M}}(y|\boldsymbol{y}_{<t}, \boldsymbol{x})$ to denote the conditional predictive distribution of generating the token at position $t$ given the prompt $\boldsymbol{x}$ and a prefix $\boldsymbol{y}_{<t} = \{y_1, y_2, ..., y_{t-1}\}$ from model $\mathcal{M}$. Then the invariance measurement function $\mathbf{V}(\tilde{\boldsymbol{x}}; \boldsymbol{x}, \mathcal{M})$ in Equation 1 is defined as follows:

$$\mathbf{V}(\tilde{\boldsymbol{x}}; \boldsymbol{x}, \mathcal{M}) = \frac{1}{N} \sum_{t=1}^{N} \mathbf{D}\Big(p_{\mathcal{M}}(y|\boldsymbol{y}_{<t}^*, \boldsymbol{x}),\ p_{\mathcal{M}}(y|\boldsymbol{y}_{<t}^*, \tilde{\boldsymbol{x}})\Big) \tag{2}$$

where $\boldsymbol{y}^* = \{y_1^*, y_2^*, ..., y_N^*\}$ is the response decoded from $\mathcal{M}$ using the original prompt $\boldsymbol{x}$, and $\mathbf{D}(\cdot, \cdot)$ is a function that quantifies the distance between two discrete probability distributions.

The final UQ score is obtained by Monte Carlo estimation of Equation 1, with $\mathbf{V}(\tilde{\boldsymbol{x}}; \boldsymbol{x}, \mathcal{M})$ substituted with Equation 2, denoted as $\mathbf{U}_{ESI}$:

$$\mathbf{U}_{ESI} := \frac{1}{L \cdot N} \sum_{l=1}^{L} \sum_{t=1}^{N} \mathbf{D}\Big(p_{\mathcal{M}}(y|\boldsymbol{y}_{<t}^*, \boldsymbol{x}),\ p_{\mathcal{M}}(y|\boldsymbol{y}_{<t}^*, \tilde{\boldsymbol{x}}_l)\Big) \tag{3}$$

The choice of function offers three advantages: First, compared to the intractable sequence distribution over the entire output space, the token distribution is fully accessible. It is also more responsive to prompt intervention, as it contains information throughout the vocabulary space, rather than focusing solely on the top-1 token. Second, measuring the difference between the same responses allows the forward process to be easily parallelized through teacher forcing, as all tokens are available in advance, thereby making it more computationally efficient compared to other methods relying on sequential generation. Third, as justified in the following Section, our approach provides a tractable approximation of epistemic-style uncertainty in LLMs.

## 3.3 ESI Estimates Epistemic Uncertainty

Epistemic Uncertainty, also known as model uncertainty, depicts the uncertainty stemming from a lack of knowledge of the underlying data-generating process (Malinin, 2019; Hüllermeier & Waegeman, 2021). Specifically, it cares about the discrepancy between the learned model $p(\boldsymbol{y}|\boldsymbol{x}, \boldsymbol{\theta})$ and the ground-truth model $p(\boldsymbol{y}|\boldsymbol{x}, \boldsymbol{\theta}^*)$, which can be quantified by

$$D_{KL}\big(p(\boldsymbol{y}|\boldsymbol{x}, \boldsymbol{\theta}) \,||\, p(\boldsymbol{y}|\boldsymbol{x}, \boldsymbol{\theta}^*)\big)$$

where $D_{KL}$ represents the Kullback–Leibler (KL) divergence.

---

[1] If $M^* > M_n$, the word $w_n$ remains unchanged.

However, for LLMs this quantity alone can be misleading: the model may guess a correct answer through spurious correlations for a particular prompt, resulting in deceptively low divergence. Instead, a model with genuinely low model uncertainty should be stable and close to the ground-truth model under all semantic preserving variants based on our analysis in Section 4.2. To capture this, we propose an alternative epistemic uncertainty estimator:

$$E_{\tilde{\boldsymbol{x}} \sim f_I(\boldsymbol{x})}\Big[D_{KL}\big(p(\boldsymbol{y}|\tilde{\boldsymbol{x}}, \boldsymbol{\theta}) \,\|\, p(\boldsymbol{y}|\boldsymbol{x}, \boldsymbol{\theta}^*)\big)\Big] \tag{4}$$

where $\tilde{\boldsymbol{x}}$ denotes the semantic-preserving variants of $\boldsymbol{x}$ sampled from distribution $f_I(\boldsymbol{x})$. The key assumption is that the ground-truth distribution remains invariant under such interventions.

However, the ground-truth model $p(\boldsymbol{y}|\boldsymbol{x}, \boldsymbol{\theta}^*)$ is intractable. Under the Bayesian setting, Malinin & Gales (2021) and Schweighofer et al. (2023) introduced the Expected Pairwise KL-Divergence (EPKL)

$$K(\boldsymbol{y}, \boldsymbol{\theta}) = E_{\boldsymbol{\theta}_1, \boldsymbol{\theta}_2}\Big[D_{KL}\big(p(\boldsymbol{y}|\boldsymbol{x}, \boldsymbol{\theta}_1)\|p(\boldsymbol{y}|\boldsymbol{x}, \boldsymbol{\theta}_2)\big)\Big]$$

as a tractable proxy, which provides a reliable approximation of the intractable epistemic uncertainty estimator:

$$K(\boldsymbol{y}, \boldsymbol{\theta}) \approx E_{\boldsymbol{\theta}}\Big[D_{KL}\big(p(\boldsymbol{y}|\boldsymbol{x}, \boldsymbol{\theta}) \,\|\, p(\boldsymbol{y}|\boldsymbol{x}, \boldsymbol{\theta}^*)\big)\Big]$$

By analogy, we extend this idea to semantic-preserving variants. Specifically, we propose to estimate equation 4 via the EPKL between the model output $\boldsymbol{y}$ and the semantic-preserving variant $\tilde{\boldsymbol{x}}$

$$K(\boldsymbol{y}, \tilde{\boldsymbol{x}}) = E_{\tilde{\boldsymbol{x}}_1, \tilde{\boldsymbol{x}}_2}\Big[D_{KL}\big(p(\boldsymbol{y}|\tilde{\boldsymbol{x}}_1, \boldsymbol{\theta})\|p(\boldsymbol{y}|\tilde{\boldsymbol{x}}_2, \boldsymbol{\theta})\big)\Big] \approx E_{\tilde{\boldsymbol{x}}}\Big[D_{KL}\big(p(\boldsymbol{y}|\tilde{\boldsymbol{x}}, \boldsymbol{\theta}) \,\|\, p(\boldsymbol{y}|\boldsymbol{x}, \boldsymbol{\theta}^*)\big)\Big]$$

where $\tilde{\boldsymbol{x}}_1, \tilde{\boldsymbol{x}}_2 \sim f_I(\boldsymbol{x})$.

We provide a theoretical justification in Appendix B that ESI (equation 3), instantiated with KL-divergence as the distance measurement function, approximates $K(\boldsymbol{y}, \tilde{\boldsymbol{x}})$, i.e.,

$$\mathbf{U}_{ESI} \approx K(\boldsymbol{y}, \tilde{\boldsymbol{x}})$$

Consequently, our proposed estimator, which quantifies the average token-wise distribution shifts across semantic-preserving interventions, serves as a tractable and reasonable proxy of equation 4, which captures epistemic-style uncertainty in LLMs.

The intuitive explanation is as follows: although the ground-truth language model is intractable, we make the assumption, based on our analysis in section 2, that the ground-truth language generation model should remain invariant under semantic-preserving interventions. Consequently, the larger the discrepancy between the learned model and the ground-truth model, the greater the average token-wise distribution variation observed in the learned model before and after the semantic-preserving intervention, which can be interpreted as a measure of epistemic uncertainty.

### 3.4 PRACTICAL CONSIDERATION

Although the KL divergence has good theoretical properties, it is not a desirable empirical distance metric, as it is non-symmetric, unbounded, and does not satisfy the triangle inequality. Therefore, we choose the Hellinger distance as the distance measure function $\mathbf{D}$ in equation 2, given by the following formula:

$$\mathbf{D}_H(p, q) = \sqrt{\frac{1}{2} \sum_i (\sqrt{p_i} - \sqrt{q_i})^2}$$

The Hellinger distance is a well-defined distance metric with values in the range $[0, 1]$, making it a more stable measure than the KL divergence. Experiment with different distance metrics can be found in Appendix D.1.

Furthermore, it is well known that the majority of the probability mass in the token predictive distribution is concentrated in a small subset of tokens, compared to the vast vocabulary space (Holtzman et al., 2020). To improve efficiency, we therefore retain only the top-$k$ most probable tokens, denoted as the truncated token predictive distribution $\tilde{p}^k(y|\boldsymbol{y}_{<t}, \boldsymbol{x})$.

Inspired by Duan et al. (2024), we employ importance weights to reduce the influence of uninformative tokens, such as the stop words. Specifically, we use a heuristic metric: the entropy of the truncated

token predictive distribution $H(\tilde{p}^k)$, based on the intuition that uninformative tokens generally exhibit low entropy (Meister et al., 2023). With the above modifications, the final implemented ESI score $\mathbf{U}^*_{ESI}$ is

$$\mathbf{U}^*_{ESI} := \frac{1}{L \cdot N} \sum_{l=1}^{L} \sum_{t=1}^{N} \boldsymbol{\alpha}_t \mathbf{D}_H \Big( \tilde{p}^k(y|\boldsymbol{y}^*_{<t}, \tilde{\boldsymbol{x}}_l), \ \tilde{p}^k(y|\boldsymbol{y}^*_{<t}, \boldsymbol{x}) \Big)$$

where $\{\tilde{\boldsymbol{x}}_l\}_{l=1}^{L}$ are the semantic-preserving variants sampled from some semantic-preserving intervention function $f_I(\boldsymbol{x})$, and $\boldsymbol{\alpha}_t$ is the entropy of truncated token predictive distribution before the intervention, i.e. $H\big(\tilde{p}^k(y|\boldsymbol{y}^*_{<t}, \boldsymbol{x})\big)$, $t = 1, 2, ..., N$.

## 4 EMPIRICAL EVALUATIONS

Following prior works (Kuhn et al., 2023; Duan et al., 2024; Abbasi-Yadkori et al., 2024), we assess the effectiveness of our Uncertainty Quantification method by predicting the correctness of model-generated answers across several widely used QA datasets, i.e., determining whether the model's outputs can be trusted.

### 4.1 EXPERIMENTAL SETTING

**Datasets.** We adopt three different types of datasets, a total of five free-form QA datasets, for our experiments. These include the open-book QA dataset CoQA (Reddy et al., 2019), where the model answers based on provided supporting documents; Two factual QA datasets with a single ground-truth answer, SciQ (Welbl et al., 2017) and TriviaQA (Joshi et al., 2017); And two QA datasets with multiple correct answers, AmbigQA (Min et al., 2020) and TruthfulQA (Lin et al., 2022), which exhibit high aleatoric uncertainty. More details please refer to Appendix C.2.

**Baselines.** We compare our proposed methods with six baselines: Length-normalized Predictive Entropy (LN-PE) (Malinin & Gales, 2021), INSIDE (Chen et al., 2024), M.I. (Abbasi-Yadkori et al., 2024), Semantic Entropy (Kuhn et al., 2023), Semantic Density (Qiu & Miikkulainen, 2024), and SAR (Duan et al., 2024). LN-PE directly uses Monte-Carlo estimation to estimate the output space entropy with length-normalized sentence log probability. INSIDE leverages the inherent variations in the semantic embeddings of sampling outputs to quantify uncertainty. M.I. assumes that multiple responses obtained from the same query should be independent and then uses the mutual information between them, which is estimated by iteratively prompting LLM, as the uncertainty score. Semantic Entropy considers the semantic equivalence and evaluates the output space entropy after clustering semantic equivalent outputs. Semantic Density applies kernel density estimation over sampled responses to reconstruct the output density, and the uncertainty score is given by the estimated density. SAR, which is the SOTA method, introduces importance weights to shift attention to more relevant tokens and sentences, thereby refining the uncertainty score.

**Models.** We utilize four base LLMs to evaluate our methods, including Llama2-chat$_{7B}$ (Touvron et al., 2023), Mistral-Nemo-Instruct$_{12B}$[2], Llama3-Instruct$_{8B}$ and Llama3-Instruct$_{70B}$ (Dubey et al., 2024). For each model, we generate original responses using greedy search and evaluate their correctness by applying the correctness metric, which serves as the correctness label. Experiments on more models are provided in Appendix E.2.

**Correctness Metric.** We employ BEM score (Bulian et al., 2022) with threshold 0.7, a semantic similarity-based correctness metric specifically developed for QA tasks, as the correctness metric rather than the Rouge-L (Lin, 2004) commonly used in prior works. The semantic-based BEM score is more reliable than the lexical overlap-based methods, as demonstrated in Kamalloo et al. (2023) and verified by our own experiments. More analysis can be found in the Appendix C.3.

**Evaluation Metric.** Following previous work (Kuhn et al., 2023; Duan et al., 2024), we use the area under the receiver operating characteristic curve (AUROC) to evaluate how effectively a UQ score predicts generation correctness. AUROC measures the score's ability to discriminate between correct and incorrect generations. An AUROC of 0.5 indicates the score is no better than random. An AUROC of 1 signifies perfect discrimination, where all UQ scores for correct generations are lower than those for incorrect ones.

---

[2]https://mistral.ai/news/mistral-nemo/

Table 1: Generation correctness prediction results, where a larger value indicates better UQ performance. Each method is evaluated 10 times on each dataset for each base model. The score outside the brackets represents the mean of the 10 trials, while the score inside the brackets indicates the standard deviation. The **bold** number represents the best performance across all methods. The underline highlights the mean value that outperforms all baselines. The asterisk ∗ indicates that the number is statistically significantly better than the SOTA baseline (SAR) at the $5\%$ significance level. The numbers in the 'Avg. Improvement' row show the average mean improvement and reduction in standard deviation compared to the SOTA baseline across models.

| Models | UQ methods | SciQ AUROC | TriviaQ AUROC | CoQA AUROC | AmbigQA AUROC | TruthfulQA AUROC |
|---|---|---|---|---|---|---|
| Llama2-chat$_{7B}$ | LN-PE | 72.10 (0.37) | 77.31 (0.14) | 66.38 (0.31) | 69.79 (0.19) | 61.63 (0.60) |
| | INSIDE | 61.79 (0.57) | 67.69 (0.14) | 62.60 (0.32) | 62.94 (0.30) | 54.48 (0.37) |
| | M.I. | 69.84 (0.43) | 75.72 (0.22) | 69.32 (0.27) | 66.96 (0.38) | 61.11 (0.88) |
| | Semantic Entropy | 71.46 (0.57) | 78.94 (0.19) | 69.72 (0.25) | 70.95 (0.40) | 56.28 (0.59) |
| | Semantic Density | 70.45 (0.56) | 76.73 (0.41) | 72.21 (0.69) | 69.72 (0.57) | 56.63 (0.86) |
| | SAR | 73.40 (0.55) | 79.45 (0.17) | 70.72 (0.39) | 70.59 (0.42) | 61.11 (0.51) |
| | Ours (SOC) | **75.30** (0.28)* | 80.22 (0.05)* | 71.69 (0.13)* | 72.29 (0.11)* | 66.82 (0.18)* |
| | Ours (Para) | 75.15 (0.22)* | **81.38** (0.07)* | **73.70** (0.09)* | **73.57** (0.12)* | **67.51** (0.22)* |
| Mistral-Instruct$_{12B}$ | LN-PE | 75.66 (0.57) | 83.66 (0.15) | 73.81 (0.24) | 74.19 (0.35) | 66.97 (0.42) |
| | INSIDE | 70.13 (0.44) | 77.99 (0.15) | 70.19 (0.23) | 68.50 (0.24) | 60.78 (0.59) |
| | M.I. | 74.16 (0.61) | 81.73 (0.18) | 75.83 (0.43) | 74.75 (0.32) | 64.61 (1.13) |
| | Semantic Entropy | 74.15 (0.58) | 84.12 (0.18) | 75.83 (0.40) | 75.88 (0.21) | 67.89 (0.66) |
| | Semantic Density | 69.66 (0.88) | 75.40 (0.70) | 75.21 (0.90) | 69.75 (1.08) | 63.01 (0.72) |
| | SAR | 77.14 (0.54) | 84.91 (0.09) | 79.58 (0.35) | 75.32 (0.43) | 68.39 (0.64) |
| | Ours (SOC) | **77.45** (0.15)* | 83.98 (0.07) | **79.81** (0.07)* | 75.24 (0.11) | **71.84** (0.24)* |
| | Ours (Para) | 77.21 (0.20) | **85.31** (0.04)* | 79.70 (0.14) | **76.99** (0.11)* | 71.01 (0.24)* |
| Llama3-Instruct$_{8B}$ | LN-PE | 71.69 (0.64) | 82.64 (0.12) | 71.11 (0.29) | 75.74 (0.37) | 65.80 (0.46) |
| | INSIDE | 60.85 (0.63) | 66.83 (0.23) | 65.35 (0.31) | 65.37 (0.57) | 61.02 (0.82) |
| | M.I. | 71.82 (0.54) | 81.31 (0.18) | 71.92 (0.25) | 74.33 (0.37) | 66.69 (0.51) |
| | Semantic Entropy | 72.49 (0.72) | 83.94 (0.12) | 71.68 (0.42) | 76.47 (0.38) | 66.54 (0.36) |
| | Semantic Density | 69.43 (0.88) | 77.09 (1.07) | 70.91 (0.44) | 69.17 (0.85) | 63.10 (0.90) |
| | SAR | 74.73 (0.45) | 84.39 (0.12) | 75.05 (0.42) | 76.58 (0.38) | 68.02 (0.66) |
| | Ours (SOC) | **75.24** (0.27)* | 84.77 (0.04)* | 77.00 (0.08)* | 79.31 (0.16)* | 68.59 (0.27)* |
| | Ours (Para) | 75.06 (0.21)* | **85.19** (0.04)* | **77.66** (0.09)* | **80.37** (0.16)* | **69.58** (0.23)* |
| Llama3-Instruct$_{70B}$ | LN-PE | 65.08 (0.70) | 72.76 (0.15) | 62.25 (0.43) | 66.88 (0.21) | 63.74 (0.98) |
| | INSIDE | 60.69 (0.42) | 61.82 (0.32) | 55.72 (0.57) | 59.05 (0.40) | 62.53 (0.54) |
| | M.I. | 62.72 (0.32) | 71.03 (0.23) | 65.47 (0.30) | 66.17 (0.38) | 61.30 (0.37) |
| | Semantic Entropy | 65.30 (0.84) | 73.12 (0.46) | 62.82 (0.55) | 68.43 (0.49) | 61.39 (0.66) |
| | Semantic Density | 62.86 (0.67) | 70.12 (0.61) | 68.31 (1.07) | 63.56 (0.64) | 59.42 (0.96) |
| | SAR | 68.59 (0.62) | 76.31 (0.16) | 68.16 (0.48) | 68.77 (0.43) | 65.08 (0.59) |
| | Ours (SOC) | 70.56 (0.10)* | 79.40 (0.04)* | 72.98 (0.08)* | 72.83 (0.16)* | 67.40 (0.14)* |
| | Ours (Para) | **71.88** (0.38)* | **80.61** (0.06)* | **75.45** (0.09)* | **74.26** (0.13)* | **69.23** (0.22)* |
| Avg. Improvement | Δ Ours(SOC) | +1.17(-0.34) | +0.83(-0.09) | +1.99(-0.32) | +2.10(-0.28) | +3.02(-0.39) |
| | Δ Ours(Para) | +1.36(-0.29) | +1.86(-0.08) | +3.25(-0.31) | +3.48(-0.29) | +3.68(-0.37) |

**Implementation Details.** As discussed in section 3.1, we implement two variations of the ESI score, each employing a different semantic-preserving intervention function: Skip-One-Char (denoted as Ours (SOC)) and Paraphrase (denoted as Ours (Para)). Paraphrases are generated using DeepSeek-V2.5 API[3]. For the SOC intervention function, we set $M^* = 3$ and $p = 0.3$. For Ours (Para), we generate five paraphrases for each input ($L = 5$), whereas for Ours (SOC), we sample ten intervened variants ($L = 10$). We retain the top-100 most probable tokens to construct the truncated token predictive distribution. For baseline methods, we adhere to the configurations specified in the respective original papers. We evaluate each method ten times across datasets and models. Additional details are provided in Appendix C.

## 4.2 RESULTS AND ANALYSIS

**Effectiveness Analysis.** Table 1 presents the main results. Both of our methods consistently outperform all baselines across most settings. In addition to superior performance, our methods

---

[3]https://api-docs.deepseek.com/news/news0905

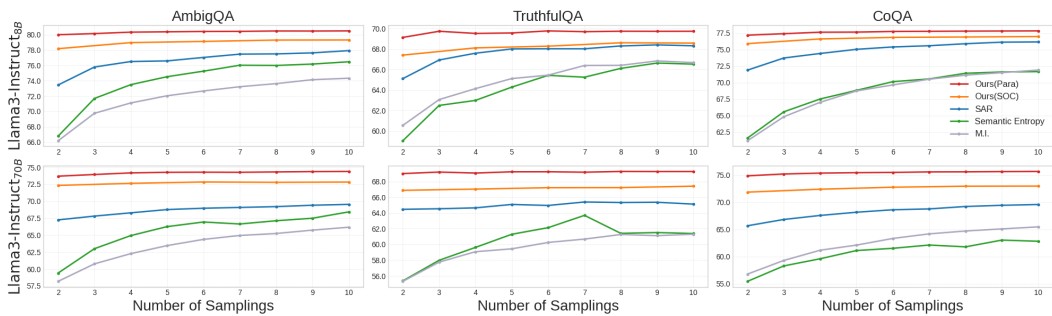

Figure 3: AUROC performance of UQ methods with different sample sizes. Our methods need fewer samples to achieve comparable performance. For ESI, the sample size corresponds to the number of intervened variants, while for baseline methods, it refers to the number of sampled generations.

Table 2: Average per-example runtime (in seconds) performed with Llama3-Instruct$_{8B}$ on a NVIDIA A100 80GB GPU. All methods are conducted with the same setting as in the main experiments.

| UQ methods | SciQ | TriviaQ | TruthfulQA |
|---|---|---|---|
| Semantic Entropy | 0.240 | 0.301 | 0.403 |
| SAR | 0.177 | 0.197 | 0.265 |
| Ours(SOC) | 0.059 | 0.067 | 0.054 |

demonstrate greater stability (i.e., lower standard deviation). This stability arises from the nature of our approach, which avoids the need to estimate variation in the intractable output space through sampling (which would lead to high variance). Instead, we focus on modeling the distribution variation of a single response before and after intervention.

Our methods achieve larger improvement on datasets exhibiting high aleatoric uncertainty (3.48 on AmbigQA and 3.68 on TruthfulQA with Para intervention), as well as on the open-book dataset (3.25 on CoQA), compared to closed-book, single-answer datasets (1.36 on SciQ and 1.86 on TriviaQA). The former improvement can be attributed to our method's ability to effectively estimate epistemic uncertainty rather than total uncertainty. In contrast, baseline methods, which estimate total uncertainty, may mistakenly attribute uncertainty arising from the data, which is inherent and normal, to erroneous generation. The enhanced performance on the open-book dataset may be due to the provided supported documents, which make the causal relationship between input and output more explicit and robust, thereby making it harder to be influenced by the intervention. As a result, correct and incorrect generations are more easily distinguishable.

**Efficiency Analysis.** As discussed in section 3.2, our method is computationally efficient, as it can leverage parallelized forward pass rather than sequential generation. We evaluate the average per-example runtime in Table 2. We observe that ESI with an efficient intervention function is 3-5 times faster than the baseline methods. Notably, although ESI with paraphrasing achieves superior performance, its efficiency is highly dependent on the size of the paraphrasing model, as noted in section 3.1. For this reason, we exclude it from the comparison in Table 2.

Additionally, Figure 3 illustrates the UQ performance across different sample sizes. It is evident that our method not only consistently outperforms the baseline methods, but also exhibits higher efficiency, in the sense that it requires a smaller number of samples (as few as 2 to 3) to achieve superior UQ performance. This efficiency arises from avoiding the need to reconstruct the intractably large output space.

### 4.3 ABLATION STUDY

**Intervention Functions.** We explore the impact of different semantic-preserving intervention functions, as illustrated in Figure 4(a). Implementation details and semantic preservation performance can be found in Appendix C.7. We can observe that all intervention methods obtain decent performance (better than SOTA) as long as the intervention efficiently preserve the semantics. However, once

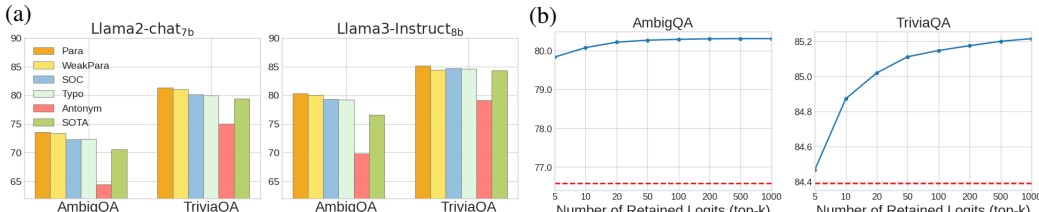

Figure 4: **(a)** AUROC performance of UQ methods with different intervention functions. 'SOTA' refers to the SOTA baseline, SAR. 'WeakPara' represents paraphrasing with a smaller model, Llama3-Instruct$_{8B}$. 'Typo' is similar to SOC, but replaces the selected character with a random one rather than skipping it. 'Antonym' indicates randomly replacing a word with its antonym. **(b)** AUROC performance of ESI (Para) with varying top-$k$ values using Llama3-Instruct$_{8B}$. The red dashed line represents the SOTA baseline performance.

the intervention hurts the semantics, our method fails because it violates our basic assumption: the response only remains invariant under semantic-equivalent variants, highlighting the importance of semantic preservation. This is shown by the large performance drop with "Antonym" intervention, which randomly replaces one word with its antonym.

Moreover, Paraphrasing-based methods (Para and WeakPara) generally outperform character-level methods (SOC and Typo), as evidenced in Table 1 and Figure 4(a). This can be attributed to the intervention intensity, where paraphrasing more effectively destroys superficial linguistic structures. Additionally, we notice a slight decrease in performance for WeakPara compared to Para, likely due to the semantic loss resulting from paraphrasing with a weaker model. This again demonstrates the importance of semantic preservation.

**Top-$k$.** We investigate the influence of the number of logits retained for the truncation predictive distribution. As shown in Figure 4(b), a monotonic trend is observed, with the curve rapidly saturating after hundreds of logits. Notably, our method yields satisfactory results even with only 5-20 logits, making it applicable to closed-source models as long as the top-$k$ logits are available. More ablation studies can be found in Appendix D

## 5 RELATED WORK

The existing approaches for uncertainty quantification (UQ) in LLMs can be broadly classified into four categories (Shorinwa et al., 2024): heuristic-based methods, including metrics such as the average log probability of the generated output (Huang et al., 2023); the verbalized methods (Mielke et al., 2022; Kadavath et al., 2022; Tian et al., 2023), where the model is directly prompted to generate an uncertainty score or to evaluate the correctness of its generated responses, with the probability of "True" being used as the uncertainty score; variation-based methods (Malinin & Gales, 2021; Kuhn et al., 2023; Lin et al., 2024; Chen et al., 2024; Qiu & Miikkulainen, 2024; Duan et al., 2024), which quantify the uncertainty by measuring the variation in the output space of responses; and test-time augmentation methods (Jiang et al., 2023; Hou et al., 2024; Abbasi-Yadkori et al., 2024), which involve deriving uncertainty scores by manipulating the input prompts. Variation-based methods have emerged as the predominant approach for UQ in LLMs, with several studies demonstrating their superiority over verbalized methods (Kuhn et al., 2023; Duan et al., 2024; Abbasi-Yadkori et al., 2024).

Our method can be categorized as a test-time augmentation method. The application of test-time augmentation methods for UQ in LLMs remains underexplored. Jiang et al. (2023) focuses on UQ of LLMs in the multiple choice tasks, rather than free-form generation, and leverages permutation-based methods to ensemble distributions over four possible choices. Hou et al. (2024) focus on quantifying the data uncertainty arising from input ambiguity. They propose an uncertainty decomposition method by introducing clarification questions, which measure the uncertainty introduced by ambiguity. Abbasi-Yadkori et al. (2024) is the most similar work to ours. Both of our works try to estimate epistemic uncertainty by making assumptions about the ground-truth language model and designing methods to measure the deviation from the assumption. However, they assume that multiple responses obtained from the same query should be independent from each other. Therefore, they use the KL-

divergence between the joint distribution and the product of marginal distributions (obtained through iteratively prompting) to measure the independence between answers. Conversely, we make the assumption that the ground-truth language model should remain invariant under semantic-preserving interventions, and quantify the average shift in the token predictive distribution of the same response before and after semantic-preserving interventions, which is not covered by previous works.

## 6 CONCLUSION

In this paper, we propose a novel approach to conduct Uncertainty Quantification for LLMs by establishing a connection between model uncertainty and invariance under semantic-preserving interventions. Our motivation stems from the basic observation that humans causally generate a response based on the semantics of the input text. Therefore, we assume that the ground-truth language generation model should be stable under semantic-equivalent interventions of input text. Our proposed method quantifies the variation in model outputs induced by such interventions, offering an effective estimate of the extent to which the model violates the assumption and, therefore, a good estimate of epistemic uncertainty. Theoretical justification supports the efficacy of our method, and extensive experiments highlight its superior performance in both effectiveness and computational efficiency. Beyond empirical gains, our causal-invariance perspective offers a new way to conduct UQ for LLMs.

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

## A    LIMITATION

Firstly, our method is a grey-box approach, which requires access to the model's output logits. Although our ablation study in Section 4.3 demonstrated that our method only needs 5-20 logits to achieve satisfactory results and most closed-source model APIs support top-$k$ log probabilities. However, these APIs recently restricted access to the log probabilities of input tokens (such as the 'echo' function previously supported by OpenAI) for safety reasons. As a result, we are unable to leverage the efficiency advantage of our method through parallelized forwarding.

Secondly, the assumption we made is that the base LLM on which our method operates is capable of generating correct answers through a robust causal pathway. This might not be true for a small model on which most of its responses (even the correct ones) are reached through spurious correlation, so our method might be inferior. Therefore, this assumption might require that the base model is well-trained, implying that it should be large and exhibit decent performance. However, this statement is not verified through extensive experiments, so we put it under limitations and wait for future investigation.

Thirdly, our study focuses on claim-level predictions (i.e., short responses), consistent with prior baselines. Performance on long-form generation may be affected due to token-wise information being spread over many correct and less informative tokens. However, claim-level evaluation remains a standard foundation for assessing longer outputs, as these are typically decomposed into individual claims (Min et al., 2023; Mohri & Hashimoto, 2024). Therefore, our method is orthogonal to long-form generation techniques and can serve as a complementary component in such pipelines.

Fourthly, our proposed estimator serves as a practical and efficient proxy for epistemic-style uncertainty in LLMs, rather than an unbiased estimator that perfectly disentangles aleatoric and epistemic uncertainty. As a result, it may be subject to approximation error and numerical instability under certain conditions. Nevertheless, our extensive empirical evaluations demonstrate that the proposed method achieves strong effectiveness and robustness across a wide range of settings, while offering substantial computational efficiency advantages.

## B    THEORETICAL JUSTIFICATION OF EPISTEMIC UNCERTAINTY APPROXIMATION

Let $\boldsymbol{x}$ be a prompt. We use $p(\boldsymbol{y}|\boldsymbol{x})$ to denote the predictive distribution of generating a sequence $\boldsymbol{y}$, and $p(y_t|\boldsymbol{y}_{<t}, \boldsymbol{x})$ to represent the conditional predictive distribution function of generating the token at position $t$ given the prompt $\boldsymbol{x}$ and a prefix $\boldsymbol{y}_{<t} = \{y_1, y_2, ..., y_{t-1}\}$. We use $\boldsymbol{y}^* \sim p(\cdot|\boldsymbol{x})$ to denote a sequence decoded from $p(\boldsymbol{y}|\boldsymbol{x})$, and $\boldsymbol{y}_{<t}^* \sim p(\cdot|\boldsymbol{x})$ to denote a subsequence with length $t-1$ decoded from $p(\boldsymbol{y}|\boldsymbol{x})$.

Without loss of generality, we define that $\boldsymbol{y}$ has a fixed length $N_{max}$ and the existence of an absorbing token $y_{eos}$, such that $p(y_{eos}|y_{t-1} = y_{eos}, \boldsymbol{y}_{<t-1}, \boldsymbol{x}) = 1$ for all $t \le N_{max}$ and $\boldsymbol{x}$. Then, we have:

$$p(\boldsymbol{y}|\boldsymbol{x}) = \prod_{t=1}^{N_{max}} p(y_t|\boldsymbol{y}_{<t}, \boldsymbol{x}) = \prod_{t=1}^{N_{eos}} p(y_t|\boldsymbol{y}_{<t}, \boldsymbol{x})$$

where $N_{eos}$ denotes the position at which the absorbing token is first generated.

Now, let us consider the EPKL between the model output $\boldsymbol{y}$ and the semantic-preserving variant $\tilde{\boldsymbol{x}}$, denoted as $K(\boldsymbol{y}, \tilde{\boldsymbol{x}})$, with the formula:

$$I_p(\boldsymbol{y}, \tilde{\boldsymbol{x}}) = E_{\tilde{\boldsymbol{x}}_1, \tilde{\boldsymbol{x}}_2}\Big[D_{KL}\big(p(\boldsymbol{y}|\boldsymbol{x}, \tilde{\boldsymbol{x}}_1)||p(\boldsymbol{y}|\boldsymbol{x}, \tilde{\boldsymbol{x}}_2)\big)\Big]$$

where $\tilde{\boldsymbol{x}}_1, \tilde{\boldsymbol{x}}_2 \sim f_I(\boldsymbol{x})$. $f_I(\boldsymbol{x})$ is a distribution over semantic-preserving variants of $\boldsymbol{x}$. We assume $P(f_I(\boldsymbol{x}) = \boldsymbol{x}) > 0$. The assumption is plausible since $\boldsymbol{x}$ is definitely semantically equivalent to itself. The derivation is as follows:

$$K(\boldsymbol{y}, \tilde{\boldsymbol{x}}) = E_{\tilde{\boldsymbol{x}}_1, \tilde{\boldsymbol{x}}_2}\Big[D_{KL}\big(p(\boldsymbol{y}|\boldsymbol{x}, \tilde{\boldsymbol{x}}_1)||p(\boldsymbol{y}|\boldsymbol{x}, \tilde{\boldsymbol{x}}_2)\big)\Big]$$

$$= E_{\tilde{\boldsymbol{x}}_1, \tilde{\boldsymbol{x}}_2}\Big[\int p(\boldsymbol{y}|\tilde{\boldsymbol{x}}_1) \log \frac{p(\boldsymbol{y}|\tilde{\boldsymbol{x}}_1)}{p(\boldsymbol{y}|\tilde{\boldsymbol{x}}_2)} d\boldsymbol{y}\Big]$$

$$= E_{\tilde{\boldsymbol{x}}_1, \tilde{\boldsymbol{x}}_2}\Big[\int p(\boldsymbol{y}|\tilde{\boldsymbol{x}}_1) \sum_{t=1}^{N_{max}} \log \frac{p(y_t|\boldsymbol{y}_{<t}, \tilde{\boldsymbol{x}}_1)}{p(y_t|\boldsymbol{y}_{<t}, \tilde{\boldsymbol{x}}_2)} d\boldsymbol{y}\Big]$$

$$= E_{\tilde{\boldsymbol{x}}_1, \tilde{\boldsymbol{x}}_2}\Big[\sum_{t=1}^{N_{max}} \int p(\boldsymbol{y}|\tilde{\boldsymbol{x}}_1) \log \frac{p(y_t|\boldsymbol{y}_{<t}, \tilde{\boldsymbol{x}}_1)}{p(y_t|\boldsymbol{y}_{<t}, \tilde{\boldsymbol{x}}_2)} d\boldsymbol{y}\Big]$$

$$= E_{\tilde{\boldsymbol{x}}_1, \tilde{\boldsymbol{x}}_2}\Big[\sum_{t=1}^{N_{max}} \int p(\boldsymbol{y}_{\le t}|\tilde{\boldsymbol{x}}_1) \log \frac{p(y_t|\boldsymbol{y}_{<t}, \tilde{\boldsymbol{x}}_1)}{p(y_t|\boldsymbol{y}_{<t}, \tilde{\boldsymbol{x}}_2)} d\boldsymbol{y}_{\le t}\Big]$$

$$= E_{\tilde{\boldsymbol{x}}_1, \tilde{\boldsymbol{x}}_2}\Big[\sum_{t=1}^{N_{max}} \int p(\boldsymbol{y}_{<t}|\tilde{\boldsymbol{x}}_1) D_{KL}\big(p(y|\boldsymbol{y}_{<t}, \tilde{\boldsymbol{x}}_1)||p(y|\boldsymbol{y}_{<t}, \tilde{\boldsymbol{x}}_2)\big) d\boldsymbol{y}_{<t}\Big]$$

$$\overset{(1)}{\approx} E_{\tilde{\boldsymbol{x}}_1, \tilde{\boldsymbol{x}}_2}\Big[\sum_{t=1}^{N_{eos}} D_{KL}\big(p(y|\boldsymbol{y}^*_{<t}, \tilde{\boldsymbol{x}}_1)||p(y|\boldsymbol{y}^*_{<t}, \tilde{\boldsymbol{x}}_2)\big)\Big], \; \boldsymbol{y}^* \sim p(\cdot|\tilde{\boldsymbol{x}}_1)$$

$$\overset{(2)}{\approx} E_{\tilde{\boldsymbol{x}}}\Big[\sum_{t=1}^{N_{eos}} D_{KL}\big(p(y|\boldsymbol{y}^*_{<t}, \boldsymbol{x})||p(y|\boldsymbol{y}^*_{<t}, \tilde{\boldsymbol{x}})\big)\Big]$$

In step (1), we approximate the intractable expectation over all prefixes:

$$\sum_{t=1}^{N_{max}} E_{\boldsymbol{y}_{<t} \sim p(\cdot|\tilde{\boldsymbol{x}}_1)}\Big[D_{KL}\big(p(y|\boldsymbol{y}_{<t}, \tilde{\boldsymbol{x}}_1)||p(y|\boldsymbol{y}_{<t}, \tilde{\boldsymbol{x}}_2)\big)\Big]$$

by a single-trajectory Monte Carlo estimation obtained from one decoded sequence $\boldsymbol{y}^* \sim p(\cdot|\tilde{\boldsymbol{x}}_1)$. This is a standard and practical approximation that preserves the structure of the underlying KL decomposition, and similar approximation has been adopted in prior work (Malinin & Gales, 2021).

In step (2), we anchor $\tilde{\boldsymbol{x}}_1$ to the original prompt $\boldsymbol{x}$, which is the specific prompt for which uncertainty is being evaluated. This step is introduced primarily for practical reasons, as it allows us to directly utilize the generated responses from the original prompt, rather than requiring the regeneration of responses using an intervened prompt. Since $\boldsymbol{x}$ is the observed and fixed prompt of interest, this operation can be viewed as a conditioning step that reduces estimator variance and improves numerical stability in practice, at the cost of introducing bias.

At last, following Malinin & Gales (2021) which considers the length-normalized 'rate', we also considered the length-normalized EPKL, $\hat{I}_p(\boldsymbol{y}, \tilde{\boldsymbol{x}})$, approximated by:

$$E_{\tilde{\boldsymbol{x}}}\Big[\frac{1}{N} \sum_{t=1}^{N} D_{KL}\big(p(y|\boldsymbol{y}^*_{<t}, \boldsymbol{x})||p(y|\boldsymbol{y}^*_{<t}, \tilde{\boldsymbol{x}})\big)\Big]$$

Table 3: Average semantic preservation scores computed between intervened and original prompts, with values ranging from 0 to 1, where 1 indicates complete semantic equivalence.

| Intervention | NLI-judge | LLM-judge |
|:---:|:---:|:---:|
| SOC | 0.989 | 0.988 |
| Para | 0.916 | 0.990 |

where $y_t^* = \arg\max p(y|\boldsymbol{y}_{<t}^*, \boldsymbol{x})$. This is exactly our ESI method with KL-divergence as the distance measurement function in equation 3. Therefore, the derivation is concluded.

## C  IMPLEMENTATION DETAILS

### C.1  SEMANTIC SIMILARITY EXPERIMENT

As discussed in section 3.1, we leverage two semantic similarity evaluation methods to examine the semantic-preserving effectiveness of intervention functions. Experiments are conducted on four datasets: SciQ, TriviaQA, AmbigQA and TruthfulQA. For each intervention function, we generate 5 samples for each query and calculate the average semantic similarity score across all datasets. The resulting semantic preservation scores are provided in Table 3.

For the NLI-judge, we utilize the Deberta-large model trained on MNLI [4]. We assign a semantic similarity score of 1 if the modified prompt is classified as entailment and 0 otherwise. For the LLM-judge, we prompt Llama3-Instruct$_{70B}$[5]. We construct a Yes/No question to prompt the LLM to check whether the intervened query is semantic equivalent to the original one. 1 is assigned if the answer contains Yes.

### C.2  DATASETS

**TriviaQA** (Joshi et al., 2017) consists of factual question-answer pairs sourced from various trivia and quiz league websites with one single ground-truth answer in semantic meaning. We evaluate on the test split for the open-domain setting with 11,313 examples.

**SciQ** (Welbl et al., 2017) comprises science exam questions covering subjects such as Chemistry, Physics, Biology, etc. Each question has one single ground-truth answer in semantic meaning. Following Duan et al. (2024), we conduct experiments on the validation split, consisting of a total of 1,000 examples.

**CoQA** (Reddy et al., 2019) is a conversational question-answering dataset with a supporting document, where all answers can be found within the provided document. Following Duan et al. (2024), we utilize the validation split, which contains a total of 7,983 examples.

**AmbigQA** (Min et al., 2020) contains questions with ambiguous meanings, leading to multiple plausible answers, thereby exhibiting high aleatoric uncertainty. We perform experiment on the validation split with a total of 2,002 examples.

**TruthfulQA** (Lin et al., 2022) consists of 817 questions across 38 categories, containing queries that some individuals might answer incorrectly due to false beliefs or misconceptions. Many of these questions are open-ended, such as "What happens to you if you eat watermelon seeds?", resulting in high aleatoric uncertainty. Experiments are performed on the entire set of 817 examples.

### C.3  CORRECTNESS METRICS

As outlined in section 4.1, we use BEM score instead of the Rouge-L to evaluate the QA correctness. In Table 4, we present the QA accuracy evaluated using three different metrics, where EM represents Exact Match, a common metric which is assigned a score of 1 if the generated response contains the entire span of the given references. Clearly, EM is a quite strict metric and should act as the lower

---

[4]deberta-large

[5]Llama-3-70B-Instruct

Table 4: QA accuracy evaluated with different correctness metrics. The threshold for BEM is set to 0.7, and the threshold for Rouge-L is 0.5

| | AmbigQA | | | TruthfulQA | | | TriviaQA | | |
| | BEM | EM | Rouge-L | BEM | EM | Rouge-L | BEM | EM | Rouge-L |
|---|---|---|---|---|---|---|---|---|---|
| Llama2-chat$_{7B}$ | 40.81 | 32.42 | 26.92 | 40.02 | 3.79 | 7.83 | 57.16 | 51.52 | 50.61 |
| Mistral-Nemo-Instruct$_{12B}$ | 44.71 | 38.16 | 40.76 | 39.90 | 5.63 | 9.18 | 67.52 | 62.43 | 65.30 |
| Llama3-Instruct$_{8B}$ | 42.81 | 35.16 | 39.36 | 31.21 | 3.43 | 5.63 | 64.67 | 58.93 | 62.64 |
| Llama3-Instruct$_{70B}$ | 60.39 | 51.55 | 55.49 | 39.29 | 6.12 | 9.06 | 80.50 | 73.79 | 78.07 |

bound for answer accuracy. However, the accuracy measured by Rouge-L is sometimes lower than that of EM, which indicates its unreliability. Moreover, both of them attain extremely low scores on TruthfulQA, which may be due to the lengthy ground-truth references. These suggest that Rouge-L is unreliable for evaluating QA performance in LLMs.

Kamalloo et al. (2023) has demonstrated that semantic-based methods, such as BEM, are more accurate than lexical overlap-based methods, such as EM and Rouge-L, through extensive experiments. For example, consider the question "What is the spiciest part of a chili pepper?" from TruthfulQA, with the ground-truth answer "The spiciest part of a chili pepper is the placenta". The model generates a correct answer "Placental tissue.", which receives a BEM score of 0.812, while both EM and Rouge-L scores are 0. Therefore, we adopt BEM as the correctness metric.

### C.4  BASELINE IMPLEMENTATION DETAILS

For Length-normalized Predictive Entropy (LN-PE) (Malinin & Gales, 2021), since the original paper uses an ensemble of models, we adopt the configuration from Kadavath et al. (2022), where ten samples are generated with a temperature of 1.0. For Semantic Entropy (Kuhn et al., 2023), we follow the newest version (Farquhar et al., 2024), which also generates ten samples with a temperature of 1.0. We utilize the Deberta-large model[6] for semantic similarity calculation. For INSIDE (Chen et al., 2024), we follow the original setting with temperature to 0.5, top-$p$ to 0.99, top-$k$ to 5, and sample 10 generations. We utilize the last token embedding in the middle layer as sentence embedding. For M.I. (Abbasi-Yadkori et al., 2024), we implement Algorithm 3 in the original paper. Following the original settings, we sample 10 responses at a temperature of 0.9 for each query and cluster answers with metric F1 (aggregate probability if F1 > 0.25). We consider the mutual information between two answers ($n = 2$), i.e., iteratively prompting LLM 2 times, and stabilization parameters $\gamma_1 = 0$ and $\gamma_2 = 0$. For Semantic Density (Qiu & Miikkulainen, 2024), we follow the original paper, which samples 10 responses with diverse beam search with diversity penalty 1.0 and beams group 10, and renormalize the token output probability with temperature 0.1. Semantic similarity (distance in their words) is evaluated with the same Deberta-large model as Semantic Entropy. For SAR (Duan et al., 2024), we follow the configuration from the original paper, which involves sampling five generations for instructed LLMs and temperature to 1.0. We utilize Cross-Encoder-Roberta-Large [7] as the original paper did.

### C.5  ADDITIONAL IMPLEMENTATION DETAILS

To construct the truncated token predictive distribution $\tilde{p}^k(y|\boldsymbol{y}_{<t}, \boldsymbol{x})$, we directly select the top-k logits from the model's output and normalize them using softmax. Notably, an issue arises when calculating the distance between two truncated token predictive distributions, as they may have different support because the top-k tokens are not identical. To address this, we expand the support of each predictive distribution to include the union of the supports of all participating distributions. Undefined logits are assigned a value equal to the minimum logit divided by 10 to smooth the distribution.

For the implementation of semantic-preserving interventions, we only intervene the queries, ensuring that the instructions in the prompt remain unchanged to preserve the model's adherence to the instructions. The intervention strategy for CoQA differs slightly because of the long documents. By

---

[6]deberta-large

[7]cross-encoder/stsb-roberta-large

default, we treat the document as part of the query for intervention. However, for the paraphrase method, we restrict the intervention to the last question only, since paraphrasing long documents is time-consuming and hard to preserve semantics.

We leverage resampling techniques to conduct repeated experiments for each method. Specifically, we first generate a large set of samples for each query, denoted as the sample size $N$, and then resample from this set multiple times to assess performance. For all baseline methods, $N = 20$. For the SOC-based ESI method, we set $N = 40$. In the case of the paraphrasing intervention method, we define a minimum value of $N = 10$ due to the diversity of paraphrases being limited by the capacity of the paraphrasing model. We prompt the paraphrase model with a maximum number of calls and retain all distinct paraphrases. If the total number of paraphrases is fewer than 10, we supplement the set with SOC-intervened queries.

## C.6   PROMPT TEMPLATES

**Template for Question Answering on QA datasets except CoQA.** We use {query} to represent the placeholder to insert the corresponding query.

> Please directly answer the following question with one or few words:
> {query}

**Template for Question Answering on CoQA.** We use {query} to represent the placeholder to insert the corresponding query. {history question} and {ground-truth answer} for conversation history since CoQA is a conversational QA dataset.

> {supported document}
> Q: {history question} A: {ground-truth answer}
> ...
>
> Please read the above article and Q&A, and directly answer the following question with one or few words:
> Q: {query} A:

**Template for semantic equivalence judgment.** We use {query} to represent the placeholder to insert the corresponding query.

> Question 1: {query1}
> Question 2: {query2}
> Please judge the semantic equivalence of the above two questions and yes means semantic equivalence. Please answer directly with no or yes:

**Template for paraphrasing.** We use {query} to represent the placeholder to insert the corresponding query. The template is inspired by Hou et al. (2024).

Table 5: Average semantic preservation scores computed between intervened and original prompts, with values ranging from 0 to 1, where 1 indicates complete semantic equivalence.

| Intervention | NLI-judge | LLM-judge |
|---|---|---|
| WeakPara | 0.717 | 0.980 |
| Typo | 0.967 | 0.932 |
| Antonym | 0.518 | 0.440 |

In this task, you will receive a single question, and your goal is to generate multiple versions of it that convey the same meaning as the original. Please format your responses as follows:
Rephrase 1: [Your rephrased question]
Rephrase 2: [Another rephrased question]
Rephrase 3: [Yet another rephrased question]
....
Ensure that each rephrased question is distinct from the others.

Here are two examples:
Question: When did the manhattan project began and end?
Rephrase 1: What were the start and end dates of the Manhattan Project?
Rephrase 2: The manhattan project began and ended in ?
Rephrase 3: What were the starting and ending dates of the Manhattan Project?
Rephrase 4: Can you tell me when the Manhattan Project started and concluded?
Rephrase 5: When was the Manhattan Project initiated and concluded?
Rephrase 6: What time period does the Manhattan Project cover, from start to finish?
Rephrase 7: Can you provide the beginning and ending dates of the Manhattan Project?

Question: Who played george washington in the john adams series?
Rephrase 1: In the John Adams series, who portrayed George Washington?
Rephrase 2: In the John Adams series, which actor portrayed George Washington?
Rephrase 3: Who portrayed George Washington in the John Adams series?
Rephrase 4: Which actor took on the role of George Washington in the John Adams series?
Rephrase 5: In the series about John Adams, who acted as George Washington?
Rephrase 6: Who was cast as George Washington in the John Adams series?
Rephrase 7: Who took on the role of George Washington in the John Adams series?

Question: {query}

## C.7 ADDITIONAL INTERVENTION FUNCTIONS IN ABLATION STUDY

In the ablation study, we implement three additional intervention functions, WeakPara, Typo, and Antonym. WeakPare shares the same implementation details as Para, except for generating the paraphrases with Llama3-Instruct$_{8B}$, a comparatively weaker model compared to DeepSeek-V2.5. Typo is a character-level method. The only difference between Typo and SOC is that SOC skips the character, while Typo replaces it with another character. The replacement is implemented with *nlpaug* package [8], which simulates the keyboard typo. Antonym randomly replaces one word in the prompts with its antonym. This method is also implemented by the *nlpaug* package.

We evaluate the semantic-preserving performance of the three additional intervention methods, as shown in Table 5. It is evident that Typo does a good job in preserving semantics, while Antonym seriously hurts it. As for WeakPara, the LLM-judge implies a perfect preservation performance, while the NLI-judge preservation score is much lower than the LLM-judge score. When compared with Para, which has an LLM-judge score of 0.990 and an NLI-judge score of 0.916, we can also observe a comparatively lower NLI-judge score. We hypothesize that the reason behind this is that

---
[8]https://nlpaug.readthedocs.io/en/latest/

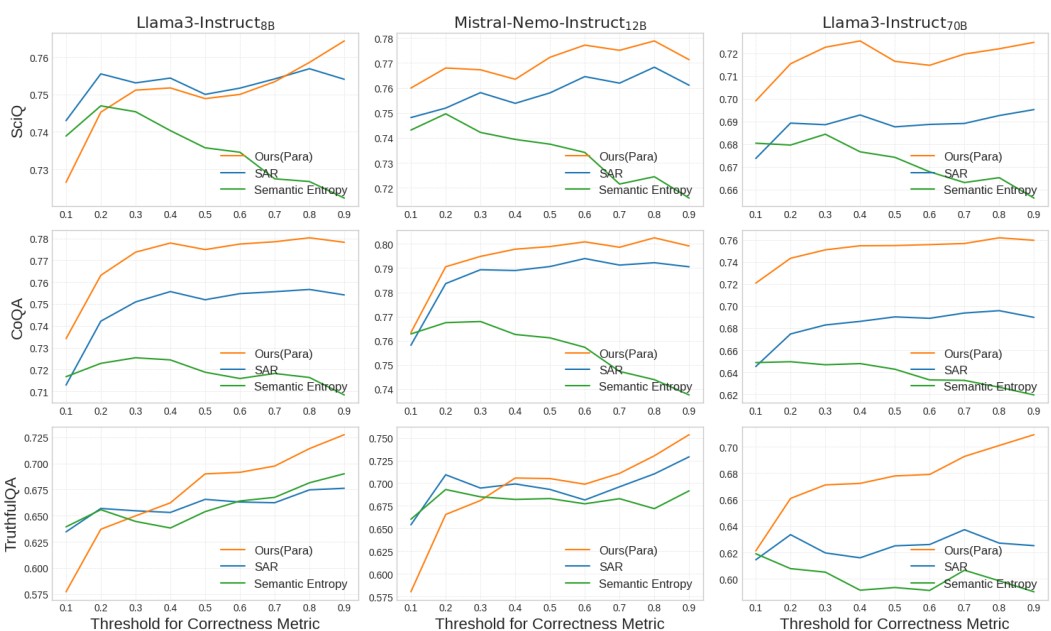

Figure 5: Performance of ESI and baseline methods with varying correctness metric threshold, with higher thresholds indicating more strict correctness criteria

Table 6: Generation correctness prediction results of our ESI (Para) method with different Distance Measuring function $D$. Scores are the average score across four base models.

|  | SciQ AUROC | TriviaQ AUROC | CoQA AUROC | AmbigQA AUROC | TruthfulQA AUROC |
|---|---|---|---|---|---|
| SAR | 73.47 | 81.27 | 73.38 | 72.82 | 65.65 |
| Bhattacharyya Distance | 72.85 | 82.20 | 76.17 | 76.39 | 67.86 |
| Square Hellinger | 74.15 | 82.76 | 77.02 | 76.62 | 68.81 |
| KL-Divergence | 73.43 | 82.13 | 76.68 | 75.66 | 67.81 |
| Hellinger Distance | 75.02 | 83.30 | 76.81 | 76.41 | 69.37 |

the NLI model suffers from some spurious correlation, which makes it unstable under paraphrasing. Therefore, we believe that WeakPara indeed suffers from some semantic loss compared with Para, but it still does a reasonable job at preserving semantics.

## D COMPLEMENTARY ABLATION STUDY

### D.1 ABLATION ON DISTANCE MEASURING FUNCTION

As shown in Table 6, Hellinger Distance exhibits the most stable performance due to its favorable properties. Notably, although the performance of other distance metrics is inferior to Hellinger, they still outperform the SOTA baseline, i.e., SAR.

### D.2 ABLATION ON CORRECTNESS METRIC THRESHOLD

As illustrated in Figure 5, our method outperforms baseline methods in most settings. We set the threshold at 0.7 to impose a more stringent correctness criterion.

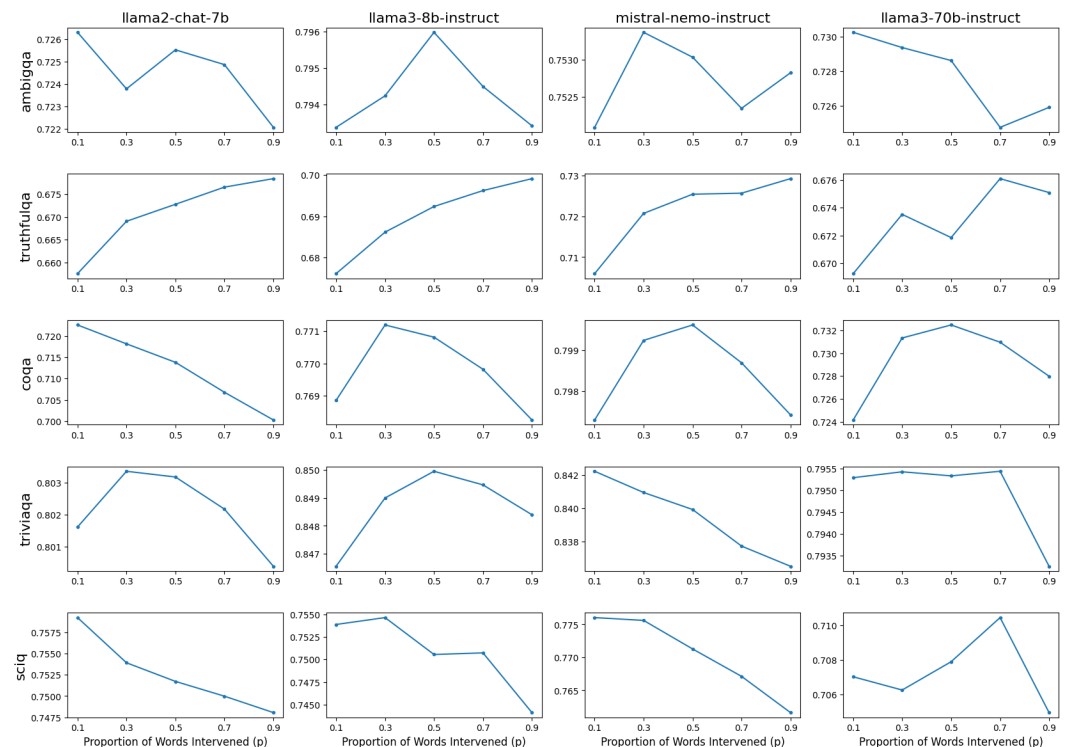

Figure 6: Performance of ESI (SOC) with varying intervention percentage $p$. A higher $p$ value indicates greater intervention intensity, resulting in poorer semantic preservation.

## D.3 ABLATION ON INTERVENTION PERCENTAGE

We evaluate our ESI (SOC) with varying hyperparameter $p$, where $p$ controls the proportion of words that are intervened by skipping one char. Higher $p$ implies a higher intervention intensity. As illustrated in Figure 6, significant declines in performance are observed across most settings, emphasizing the importance of semantic preservation. The only exception is the TruthfulQA dataset, which benefits from a higher intervention intensity. We hypothesize that this is due to the nature of TruthfulQA, which includes questions that individuals may answer incorrectly due to false beliefs or misconceptions. This suggests that the correlation between input and some erroneous answers might also be causal and could be learned from the data. To distinguish these from the more robust correct causal pathways, stronger intervention is required to destroy the incorrect causal links.

## E COMPLEMENTARY EXPERIMENTS

### E.1 COMPLEMENTARY EXPERIMENTS ON MORE BASELINE METHODS

We compare our method with several black-box UQ methods, which do not have access to model logits, and P(True) (Kadavath et al., 2022),which is the most popular verbalized method, as shown in Table 7. Empirical Entropy (Kuhn et al., 2023) computes the entropy of the empirical distribution of the semantic-clustered answers. Self-Con refers to self-consistency (Wang et al., 2023), which calculates the UQ score based on the proportion of responses that are semantically equivalent to the greedy-generated response among all sampled responses. We utilize the Deberta-large model[9] for semantic similarity calculation. Self-Con (DegMat) (Lin et al., 2024) leverages the spectral clustering method to transform the similarity matrix between different sampled responses into uncertainty scores. DegMat stands for the Degree Matrix, and the resulting UQ is actually the average of all pair-wise similarity scores. The similarity estimation function used is identical to that of the self-consistency

---

[9]deberta-large

Table 7: Generation correctness prediction results, where a larger value indicates better UQ performance.

| Models | UQ methods | SciQ AUROC | TriviaQ AUROC | CoQA AUROC | AmbigQA AUROC | TruthfulQA AUROC |
|---|---|---|---|---|---|---|
| Llama2-chat$_{7B}$ | Empirical Entropy | 70.06 | 78.61 | 69.21 | 70.05 | 55.84 |
| | Self-Con | 70.09 | 75.83 | 67.85 | 67.89 | 52.74 |
| | Self-Con (DegMat) | 71.99 | 76.84 | 72.09 | 69.01 | 53.52 |
| | ICE | 72.21 | 82.74 | 73.90 | 73.15 | 58.87 |
| | P(True) | 69.23 | 71.97 | 48.84 | 65.83 | 56.88 |
| | SAR | 73.40 | 79.45 | 70.72 | 70.59 | 61.11 |
| | Ours (SOC) | 75.30 | 80.22 | 71.69 | 72.29 | 66.82 |
| | Ours (Para) | 75.15 | 81.38 | 73.70 | 73.57 | 67.51 |
| Mistral-Nemo-Instruct$_{12B}$ | Empirical Entropy | 71.93 | 83.61 | 75.30 | 75.29 | 68.29 |
| | Self-Con | 72.83 | 81.33 | 75.69 | 74.28 | 65.38 |
| | Self-Con (DegMat) | 75.31 | 82.75 | 80.69 | 75.22 | 66.65 |
| | ICE | 74.88 | 85.91 | 76.32 | 77.00 | 69.66 |
| | P(True) | 71.42 | 81.35 | 57.16 | 71.37 | 53.86 |
| | SAR | 77.14 | 84.91 | 79.58 | 75.32 | 68.39 |
| | Ours (SOC) | 77.45 | 83.98 | 79.81 | 75.24 | 71.84 |
| | Ours (Para) | 77.21 | 85.31 | 79.70 | 76.99 | 71.01 |
| Llama3-Instruct$_{8B}$ | Empirical Entropy | 73.06 | 83.29 | 71.85 | 76.83 | 67.25 |
| | Self-Con | 69.73 | 81.43 | 68.88 | 74.52 | 64.25 |
| | Self-Con (DegMat) | 72.42 | 82.73 | 74.73 | 75.91 | 64.38 |
| | ICE | 73.93 | 85.74 | 75.04 | 77.03 | 71.77 |
| | P(True) | 57.40 | 73.81 | 43.73 | 70.48 | 51.92 |
| | SAR | 74.73 | 84.39 | 75.05 | 76.58 | 68.02 |
| | Ours (SOC) | 75.24 | 84.77 | 77.00 | 79.31 | 68.59 |
| | Ours (Para) | 75.06 | 85.19 | 77.66 | 80.37 | 69.58 |
| Llama3-Instruct$_{70B}$ | Empirical Entropy | 64.09 | 72.60 | 63.15 | 67.40 | 61.76 |
| | Self-Con | 60.51 | 70.06 | 64.52 | 66.12 | 59.01 |
| | Self-Con (DegMat) | 62.54 | 73.47 | 73.27 | 67.41 | 61.70 |
| | ICE | 69.82 | 80.04 | 72.88 | 71.72 | 67.07 |
| | P(True) | 65.62 | 75.61 | 42.95 | 70.15 | 58.04 |
| | SAR | 68.59 | 76.31 | 68.16 | 68.77 | 65.08 |
| | Ours (SOC) | 70.56 | 79.40 | 72.98 | 72.83 | 67.40 |
| | Ours (Para) | 71.88 | 80.61 | 75.45 | 74.26 | 69.23 |

method. P(True) (Kadavath et al., 2022) directly prompts the LLM with a True/False question to evaluate the correctness of its generated responses without ground-truth. The probability of "True" being generated is used as the uncertainty score.

It is worth noting that ICE is the method used by Hou et al. (2024) on their experiments on quantifying total uncertainty. While their primary focus is on generating input clarification questions to quantify aleatoric uncertainty, they paraphrase the input queries and quantify the output space variation by the ensemble method for this particular experiment. Specifically, they generate ten paraphrases for each query and sample ten responses for each paraphrased query. They then calculate the empirical entropy for each paraphrased query and ensemble the ten empirical distributions by simply averaging them. To enable the averaging, they have to cluster the 100 responses together. For cost consideration, we utilize the Deberta-large model in the same way as self-consistency did to conduct the clustering. Nevertheless, the computational cost is still high, we only perform the experiment once for each black-box method. For SAR and our method, we still report the average performance of 10 trials,

As presented in Table 7, our method consistently outperforms the baselines in most settings, with the exception of ICE on TriviaQA. This can be attributed to TriviaQA having low aleatoric uncertainty, which means our epistemic uncertainty estimation method offers no significant advantage. However, our method still achieves comparable results with significantly lower computational costs.

Table 8: Generation correctness prediction results, where a larger value indicates better UQ performance. Each method is evaluated 10 times on each dataset for each base model. The score outside the brackets represents the mean of the 10 trials, while the score inside the brackets indicates the standard deviation. The **bold** number represents the best performance across all methods. The underline highlights the mean value that outperforms all baselines. The asterisk ∗ indicates that the number is statistically significantly better than the SOTA baseline (SAR) at the $5\%$ significance level.

| Models | UQ methods | SciQ AUROC | TriviaQ AUROC | CoQA AUROC | AmbigQA AUROC | TruthfulQA AUROC |
|---|---|---|---|---|---|---|
| Llama3.1-Instruct$_{8B}$ | LN-PE | 72.02 (0.74) | 84.55 (0.10) | 78.33 (0.45) | 78.60 (0.32) | 63.55 (0.43) |
| | INSIDE | 68.54 (0.81) | 76.14 (0.14) | 73.27 (0.18) | 72.56 (0.51) | 55.48 (0.63) |
| | M.I. | 72.37 (0.46) | 83.22 (0.10) | 78.30 (0.39) | 75.27 (0.17) | 64.66 (0.61) |
| | Semantic Entropy | 69.75 (0.78) | 83.65 (0.24) | 76.77 (0.29) | 78.16 (0.22) | 62.84 (0.94) |
| | Semantic Density | 71.96 (0.73) | 81.37 (0.31) | 77.30 (1.16) | 74.81 (0.50) | 58.59 (1.52) |
| | SAR | 74.43 (0.56) | 85.86 (0.11) | 81.52 (0.46) | 79.01 (0.36) | 63.79 (0.52) |
| | Ours(SOC) | 75.30 (0.24)∗ | 85.02 (0.04) | 82.24 (0.06)∗ | 78.87 (0.12) | 64.56 (0.34)∗ |
| | Ours(Para) | **74.95** (0.22)∗ | **86.07** (0.05)∗ | **82.38** (0.10)∗ | **80.61** (0.15)∗ | **65.35** (0.26)∗ |
| Qwen2.5-Instruct$_{14B}$ | LN-PE | 70.55 (0.29) | 81.69 (0.10) | 65.24 (0.21) | 72.67 (0.15) | 65.09 (0.44) |
| | INSIDE | 60.29 (0.42) | 69.78 (0.20) | 63.77 (0.31) | 62.25 (0.29) | 61.55 (0.74) |
| | M.I. | 64.33 (0.53) | 76.28 (0.25) | 62.98 (0.27) | 70.63 (0.32) | 61.42 (1.07) |
| | Semantic Entropy | 54.72 (1.19) | 73.66 (0.21) | 57.14 (0.46) | 71.47 (0.43) | 58.23 (0.69) |
| | Semantic Density | 66.00 (0.55) | 74.01 (0.25) | **73.77** (0.57) | 67.78 (0.39) | 59.88 (1.02) |
| | SAR | 70.76 (0.52) | 81.69 (0.14) | 69.31 (0.20) | 72.62 (0.19) | 65.13 (0.59) |
| | Ours(SOC) | **71.88** (0.17)∗ | 83.33 (0.04)∗ | 66.78 (0.14) | 74.57 (0.09)∗ | 65.48 (0.15)∗ |
| | Ours(Para) | 71.68 (0.11)∗ | **84.11** (0.02)∗ | 69.00 (0.09) | **75.34** (0.07)∗ | **65.75** (0.12)∗ |
| Qwen3-Instruct$_{4B}$ | LN-PE | 72.45 (0.46) | 81.32 (0.07) | 64.67 (0.19) | 74.36 (0.27) | 66.22 (0.55) |
| | INSIDE | 58.69 (0.56) | 72.21 (0.16) | 56.30 (0.35) | 66.84 (0.30) | 63.00 (0.64) |
| | M.I. | 65.60 (0.74) | 76.28 (0.08) | 59.71 (0.36) | 73.19 (0.23) | 64.08 (0.36) |
| | Semantic Entropy | 58.24 (0.93) | 76.15 (0.15) | 53.39 (0.53) | 73.34 (0.27) | 61.55 (0.77) |
| | Semantic Density | 63.51 (0.65) | 76.00 (0.43) | **70.97** (0.76) | 72.42 (0.41) | 58.75 (0.71) |
| | SAR | 72.65 (0.37) | 81.43 (0.11) | 68.70 (0.23) | 74.49 (0.17) | 67.87 (0.36) |
| | Ours(SOC) | 74.23 (0.13)∗ | 82.04 (0.03)∗ | 66.67 (0.04) | 75.33 (0.09)∗ | **69.27** (0.14)∗ |
| | Ours(Para) | **74.44** (0.12)∗ | **83.08** (0.03)∗ | 69.41 (0.10)∗ | **76.53** (0.07)∗ | 68.96 (0.23)∗ |

### E.2 COMPLEMENTARY EXPERIMENTS ON MORE MODELS

We conduct main experiments on three additional recent models, Llama3.1-Instruct$_{8B}$[10], Qwen2.5-Instruct$_{14B}$(Qwen et al., 2025) and Qwen3-Instruct$_{4B}$(Yang et al., 2025), as shown in Table 8. Our method still outperforms all baseline across most settings, which demonstrate the robustness and effectiveness of our method.

### E.3 COMPLEMENTARY EXPERIMENTS ON MORE EVALUATION METRICS

We report additional experiments using several commonly used evaluation metrics, including AURC (Area Under the Risk–Coverage Curve, Table 9), AUCPR (Area Under the Precision–Recall Curve, Table 10), and selective accuracy at fixed coverages (Acc@80% in Table 11 and Acc@90% in Table 12).

Our method consistently outperforms baselines in terms of AUROC and AURC, indicating that it provides a more reliable global ranking of examples by uncertainty and achieves lower overall risk across coverage levels. For AUCPR, our method generally outperforms most baselines, but the results are less stable on TriviaQA and CoQA. This behavior is consistent with the known sensitivity of AUCPR under strong class imbalance, which amplifies the effect of small ranking differences in highly skewed datasets. In particular, the strong generation performance (as shown in Table 12) of baseline models on these two datasets lead to an imbalance of positive labels.

For selective accuracy at fixed coverages (Acc@80% and Acc@90%), the results are more model and dataset dependent. Our methods show strong and consistent improvements on Llama2-Chat-7B and LLama3-Instruct-70B at Acc@80%, while exhibiting more variable behavior on other backbones, where the performance differences are generally small and comparable. Our methods also tend to be more robust on datasets with high aleatoric uncertainty, such as AmbigQA and TruthfulQA,

---

[10]https://ai.meta.com/blog/meta-llama-3-1/

Table 9: Generation correctness prediction results evaluated with AURC, where a smaller value indicates better UQ performance. Each method is evaluated 10 times on each dataset for each base model. The score outside the brackets represents the mean of the 10 trials, while the score inside the brackets indicates the standard deviation. The **bold** number represents the best performance across all methods. The underline highlights the mean value that outperforms all baselines. The asterisk $*$ indicates that the number is statistically significantly better than the SOTA baseline (SAR) at the $5\%$ significance level. The numbers in the 'Avg. Improvement' row show the average mean improvement and reduction in standard deviation compared to the SOTA baseline across models.

| Models | UQ methods | SciQ AURC | TriviaQ AURC | CoQA AURC | AmbigQA AURC | TruthfulQA AURC |
|---|---|---|---|---|---|---|
| Llama2-chat$_{7B}$ | LN-PE | 36.06 (0.46) | 23.76 (0.07) | 15.85 (0.08) | 44.83 (0.29) | 52.69 (0.42) |
| | INSIDE | 45.84 (0.87) | 33.64 (0.29) | 18.27 (0.40) | 51.37 (0.43) | 55.21 (0.92) |
| | M.I. | 38.14 (0.97) | 26.36 (0.23) | 15.27 (0.15) | 47.92 (0.89) | 51.34 (1.21) |
| | Semantic Entropy | 37.74 (1.06) | 24.03 (0.24) | 14.64 (0.26) | 44.13 (0.68) | 54.46 (1.48) |
| | Semantic Density | 36.45 (0.40) | 24.58 (0.53) | **12.17** (0.14) | 45.07 (0.22) | 54.20 (0.83) |
| | SAR | 34.57 (0.46) | 22.22 (0.10) | 14.04 (0.17) | 43.78 (0.28) | 52.59 (0.50) |
| | Ours(SOC) | 32.52 (0.16)* | 21.32 (0.05)* | 13.35 (0.05)* | 42.11 (0.09)* | 50.24 (0.18)* |
| | Ours(Para) | 32.99 (0.22)* | **20.66** (0.06)* | 12.23 (0.04)* | **41.57** (0.21)* | **49.42** (0.37)* |
| Mistral-Instruct$_{12B}$ | LN-PE | 24.38 (0.37) | 12.66 (0.06) | 6.52 (0.09) | 37.53 (0.52) | 48.12 (0.31) |
| | INSIDE | 31.85 (0.54) | 18.05 (0.15) | 8.03 (0.09) | 43.20 (0.27) | 52.46 (0.94) |
| | M.I. | 26.78 (0.66) | 15.07 (0.40) | 6.79 (0.21) | 38.70 (0.36) | 48.98 (1.41) |
| | Semantic Entropy | 27.90 (0.65) | 13.99 (0.16) | 6.83 (0.17) | 36.79 (0.33) | 47.64 (0.53) |
| | Semantic Density | 29.59 (0.78) | 18.62 (0.39) | 6.02 (0.16) | 41.17 (0.54) | 49.58 (0.81) |
| | SAR | 23.37 (0.47) | 12.18 (0.04) | 5.22 (0.05) | 36.96 (0.35) | 47.22 (0.32) |
| | Ours(SOC) | 22.58 (0.14)* | 12.38 (0.02) | 4.89 (0.03)* | 36.21 (0.08)* | 42.94 (0.17)* |
| | Ours(Para) | **22.44** (0.14)* | **11.67** (0.02)* | **4.76** (0.04)* | **34.92** (0.06)* | **42.63** (0.28)* |
| Llama3-Instruct$_{8B}$ | LN-PE | 28.65 (0.54) | 16.04 (0.07) | 9.66 (0.12) | 39.59 (0.48) | 58.23 (0.45) |
| | INSIDE | 35.74 (0.48) | 26.02 (0.15) | 11.78 (0.17) | 46.49 (0.53) | 59.33 (0.71) |
| | M.I. | 29.94 (0.71) | 17.27 (0.28) | 9.72 (0.19) | 41.29 (0.48) | 57.95 (1.48) |
| | Semantic Entropy | 30.22 (0.94) | 15.93 (0.15) | 10.10 (0.35) | 39.36 (0.49) | 57.98 (1.37) |
| | Semantic Density | 30.52 (0.78) | 19.38 (0.64) | 8.80 (0.22) | 43.13 (0.95) | 63.74 (0.63) |
| | SAR | 26.70 (0.35) | 14.88 (0.07) | 8.39 (0.15) | 39.13 (0.37) | 57.05 (0.55) |
| | Ours(SOC) | 25.22 (0.16)* | 14.40 (0.04)* | 7.72 (0.03)* | 35.94 (0.12)* | 55.70 (0.20)* |
| | Ours(Para) | 25.30 (0.16)* | **14.05** (0.06)* | **7.08** (0.04)* | **35.61** (0.14)* | **54.71** (0.28)* |
| Llama3-Instruct$_{70B}$ | LN-PE | 24.82 (0.34) | 11.27 (0.12) | 10.77 (0.11) | 29.21 (0.21) | 52.53 (0.39) |
| | INSIDE | 26.44 (0.57) | 14.08 (0.15) | 12.64 (0.16) | 32.57 (0.42) | 52.72 (0.44) |
| | M.I. | 26.87 (0.56) | 12.61 (0.11) | 10.04 (0.28) | 31.67 (0.70) | 54.75 (1.83) |
| | Semantic Entropy | 25.21 (0.57) | 11.61 (0.19) | 10.94 (0.35) | 28.86 (0.64) | 54.03 (1.19) |
| | Semantic Density | 26.99 (0.24) | 12.15 (0.23) | 7.96 (0.13) | 30.48 (0.54) | 56.89 (0.49) |
| | SAR | 23.07 (0.51) | 10.05 (0.07) | 8.99 (0.09) | 29.03 (0.31) | 51.06 (0.69) |
| | Ours(SOC) | 21.80 (0.16)* | 8.58 (0.02)* | 7.56 (0.04)* | 24.94 (0.10)* | 50.21 (0.11)* |
| | Ours(Para) | **20.35** (0.18)* | **8.02** (0.07)* | **6.35** (0.04)* | **24.03** (0.12)* | **47.29** (0.23)* |

while showing greater variability on other datasets. A similar trend is observed for Acc@90%. Despite these local variations, the consistently strong AURC performance indicates that the proposed method provides more reliable global uncertainty rankings, while remaining competitive at specific high-coverage operating points.

In summary, our methods demonstrate consistently superior performance on datasets with higher inherent aleatoric uncertainty across all evaluation metrics, providing strong empirical evidence in support of our motivation of estimating an epistemic-style uncertainty. On datasets with more deterministic answer patterns, our method still achieves stronger overall performance in terms of AUROC, AURC, and AUCPR, while exhibiting some variability (still comparative) in local metrics as measured by selective accuracy at fixed coverages. These local fluctuations are generally small in magnitude and do not affect the overall robustness of the proposed method.

### E.4 COMPLEMENTARY EXPERIMENTS ON COMPUTATION EFFICIENCY

As stated in Section 4.2, the efficiency of paraphrasing is highly dependent on the size and deployment of the paraphrasing model. To provide a clearer picture, we report the average per-example runtime

Table 10: Generation correctness prediction results evaluated with AUCPR, where a larger value indicates better UQ performance. Each method is evaluated 10 times on each dataset for each base model. The score outside the brackets represents the mean of the 10 trials, while the score inside the brackets indicates the standard deviation. The **bold** number represents the best performance across all methods. The underline highlights the mean value that outperforms all baselines. The asterisk $*$ indicates that the number is statistically significantly better than the SOTA baseline (SAR) at the $5\%$ significance level. The numbers in the 'Avg. Improvement' row show the average mean improvement and reduction in standard deviation compared to the SOTA baseline across models.

| Models | UQ methods | SciQ AUCPR | TriviaQ AUCPR | CoQA AUCPR | AmbigQA AUCPR | TruthfulQA AUCPR |
|---|---|---|---|---|---|---|
| Llama2-chat$_{7B}$ | LN-PE | 72.12 (0.81) | 70.85 (0.15) | 39.00 (0.63) | 76.49 (0.35) | 71.76 (0.75) |
| | INSIDE | 63.30 (0.50) | 63.25 (0.27) | 37.47 (0.26) | 70.23 (0.48) | 63.91 (0.57) |
| | M.I. | 69.23 (0.49) | 71.18 (0.20) | 44.07 (0.39) | 74.18 (0.33) | 70.05 (0.73) |
| | Semantic Entropy | 73.20 (0.64) | 73.64 (0.15) | 44.85 (0.39) | 76.09 (0.32) | 61.78 (0.43) |
| | Semantic Density | 71.15 (0.34) | 70.30 (0.42) | 41.79 (0.86) | 74.97 (0.55) | 63.69 (0.99) |
| | SAR | 73.17 (0.64) | 73.02 (0.15) | 45.42 (0.79) | 76.32 (0.38) | 70.33 (0.56) |
| | Ours(SOC) | 75.45 (0.24)* | 74.68 (0.12)* | 47.40 (0.44)* | 78.88 (0.18)* | 77.28 (0.38)* |
| | Ours(Para) | **76.30** (0.29)* | **75.99** (0.08)* | **49.02** (0.08)* | **80.15** (0.09)* | **78.00** (0.28)* |
| Mistral-Instruct$_{12B}$ | LN-PE | 68.64 (0.32) | 70.62 (0.21) | 35.48 (0.67) | 78.51 (0.43) | 73.47 (0.64) |
| | INSIDE | 62.25 (0.81) | 65.82 (0.23) | 32.74 (0.45) | 71.89 (0.39) | 68.00 (0.73) |
| | M.I. | 67.14 (1.00) | 70.32 (0.30) | 37.81 (0.69) | 79.13 (0.38) | 71.03 (0.72) |
| | Semantic Entropy | 69.20 (0.57) | **73.92** (0.14) | 42.42 (0.40) | 78.76 (0.37) | 74.67 (0.62) |
| | Semantic Density | 63.90 (1.44) | 62.93 (0.65) | 35.37 (1.37) | 74.34 (0.79) | 70.26 (0.59) |
| | SAR | 70.26 (0.97) | 72.42 (0.32) | **43.53** (0.70) | 79.19 (0.40) | 74.71 (0.93) |
| | Ours(SOC) | **70.57** (0.23) | 70.74 (0.08) | 40.65 (0.25) | 78.49 (0.24) | **75.88** (0.36)* |
| | Ours(Para) | 70.51 (0.43) | 72.29 (0.06) | 39.30 (0.20) | **80.92** (0.14)* | 74.46 (0.30) |
| Llama3-Instruct$_{8B}$ | LN-PE | 65.11 (0.87) | 73.74 (0.24) | 37.15 (0.52) | 81.13 (0.39) | 80.98 (0.61) |
| | INSIDE | 55.70 (0.61) | 58.74 (0.21) | 33.56 (0.43) | 72.64 (0.44) | 75.67 (0.36) |
| | M.I. | 67.11 (0.57) | 72.73 (0.23) | 40.15 (0.86) | 79.99 (0.36) | 81.08 (0.48) |
| | Semantic Entropy | 69.33 (0.59) | **76.79** (0.17) | 42.63 (0.39) | 81.44 (0.32) | 80.93 (0.62) |
| | Semantic Density | 64.33 (1.38) | 66.80 (0.79) | 34.06 (1.65) | 75.82 (0.86) | 79.48 (0.93) |
| | SAR | 68.90 (0.80) | 75.91 (0.24) | 45.45 (0.99) | 81.55 (0.48) | 82.01 (0.63) |
| | Ours(SOC) | **69.82** (0.20)* | 76.03 (0.06) | 45.29 (0.17) | 83.50 (0.20)* | 82.25 (0.20) |
| | Ours(Para) | 69.60 (0.36) | 75.99 (0.13) | **45.72** (0.07) | **85.03** (0.07)* | **82.30** (0.19)* |
| Llama3-Instruct$_{70B}$ | LN-PE | 48.89 (0.78) | 49.29 (0.45) | 25.53 (0.44) | 60.68 (0.65) | 72.75 (0.69) |
| | INSIDE | 42.05 (0.93) | 34.96 (0.45) | 21.89 (0.45) | 51.90 (0.69) | 69.87 (0.61) |
| | M.I. | 53.67 (0.76) | 53.52 (0.48) | 35.38 (0.40) | 61.18 (0.51) | 73.01 (0.53) |
| | Semantic Entropy | 54.61 (0.71) | 55.78 (0.37) | 29.85 (0.43) | 62.63 (0.28) | 71.02 (0.30) |
| | Semantic Density | 52.26 (0.66) | 44.10 (1.18) | 25.85 (0.82) | 55.23 (1.27) | 69.54 (0.43) |
| | SAR | 56.51 (0.84) | 55.24 (0.36) | 33.86 (0.47) | 62.57 (0.39) | 73.60 (0.76) |
| | Ours(SOC) | 57.35 (0.16)* | 57.80 (0.14)* | 34.06 (0.11) | 67.17 (0.11)* | 75.26 (0.28)* |
| | Ours(Para) | 56.21 (0.37) | **58.67** (0.19)* | **36.00** (0.13)* | **68.25** (0.24)* | **75.62** (0.34)* |

of two paraphrasing implementations, as shown in Table 13 (Para implemented using the DeepSeek API and WeakPara implemented using Llama3-Instruct$_{8B}$).

It is worth noting that the latency of API-based methods is affected by external factors such as network conditions and server load, and thus does not necessarily reflect the intrinsic model speed. The paraphrasing speed of Llama3-Instruct$_{8B}$ could potentially be further improved through engineering optimizations (e.g., alternative decoding strategies or shorter prompts), but such system-level optimizations are beyond the scope of this paper.

# F  THE USE OF LARGE LANGUAGE MODELS

We employed LLMs to polish certain sections of our writing and to generate routine, non-novel code.

Table 11: Generation correctness prediction results evaluated with Selective Accuracy at $80\%$ Coverage (Acc@80), where a larger value indicates better UQ performance. Each method is evaluated 10 times on each dataset for each base model. The score outside the brackets represents the mean of the 10 trials, while the score inside the brackets indicates the standard deviation. The **bold** number represents the best performance across all methods. The underline highlights the mean value that outperforms all baselines. The asterisk $*$ indicates that the number is statistically significantly better than the SOTA baseline (SAR) at the $5\%$ significance level. The numbers in the 'Avg. Improvement' row show the average mean improvement and reduction in standard deviation compared to the SOTA baseline across models.

| Models | UQ methods | SciQ Acc@80 | TriviaQ Acc@80 | CoQA Acc@80 | AmbigQA Acc@80 | TruthfulQA Acc@80 |
|---|---|---|---|---|---|---|
| Llama2-chat$_{7B}$ | LN-PE | 55.11 (0.48) | 65.75 (0.11) | 81.79 (0.17) | 46.77 (0.19) | 44.46 (0.55) |
| | INSIDE | 53.39 (0.21) | 64.32 (0.09) | 81.78 (0.13) | 45.32 (0.30) | 40.03 (0.40) |
| | M.I. | 53.80 (0.41) | 65.42 (0.06) | 82.79 (0.14) | 45.66 (0.25) | 43.54 (0.52) |
| | Semantic Entropy | 55.56 (0.32) | 66.55 (0.09) | 83.27 (0.19) | 46.81 (0.20) | 40.47 (0.23) |
| | Semantic Density | 54.54 (0.17) | 65.68 (0.16) | 82.36 (0.21) | 45.96 (0.32) | 41.01 (0.94) |
| | SAR | 55.11 (0.39) | 66.26 (0.11) | 83.41 (0.17) | 46.51 (0.27) | 43.45 (0.42) |
| | Ours(SOC) | 56.53 (0.24)* | 66.65 (0.04)* | 83.38 (0.11) | 47.51 (0.18)* | 46.28 (0.29)* |
| | Ours(Para) | **56.95** (0.15)* | **66.87** (0.10)* | **83.77** (0.06)* | **48.01** (0.06)* | **47.47** (0.26)* |
| Mistral-Instruct$_{12B}$ | LN-PE | 65.83 (0.36) | 78.10 (0.16) | 90.74 (0.12) | 52.34 (0.12) | 44.70 (0.58) |
| | INSIDE | 64.53 (0.33) | 77.61 (0.09) | 90.43 (0.07) | 50.80 (0.16) | 43.11 (0.25) |
| | M.I. | 64.78 (0.43) | 77.54 (0.08) | 91.61 (0.17) | 52.73 (0.19) | 43.25 (0.74) |
| | Semantic Entropy | **66.09** (0.37) | **79.17** (0.09) | **92.16** (0.07) | 52.37 (0.27) | **45.08** (0.42) |
| | Semantic Density | 64.19 (0.45) | 76.75 (0.31) | 90.25 (0.56) | 51.28 (0.38) | 43.31 (0.26) |
| | SAR | 65.95 (0.40) | 78.66 (0.12) | 92.11 (0.09) | 52.42 (0.27) | 45.01 (0.44) |
| | Ours(SOC) | 65.94 (0.25) | 77.95 (0.08) | 92.01 (0.07) | 52.56 (0.20) | 44.47 (0.21) |
| | Ours(Para) | 65.89 (0.33) | 78.27 (0.07) | 91.46 (0.03) | 53.10 (0.10)* | 44.41 (0.22) |
| Llama3-Instruct$_{8B}$ | LN-PE | 63.15 (0.17) | 75.61 (0.08) | 88.16 (0.14) | 50.69 (0.20) | 35.90 (0.16) |
| | INSIDE | 60.44 (0.37) | 72.62 (0.12) | 87.48 (0.11) | 48.86 (0.28) | 33.45 (0.31) |
| | M.I. | 63.40 (0.27) | 74.85 (0.06) | 88.98 (0.12) | 50.41 (0.27) | 35.45 (0.38) |
| | Semantic Entropy | **64.69** (0.34) | **76.57** (0.06) | **89.67** (0.09) | 50.90 (0.30) | 35.80 (0.34) |
| | Semantic Density | 63.41 (0.60) | 73.99 (0.50) | 86.41 (0.34) | 49.66 (0.31) | 35.42 (0.51) |
| | SAR | 64.34 (0.53) | 76.14 (0.12) | 89.61 (0.20) | 50.97 (0.20) | 35.68 (0.41) |
| | Ours(SOC) | 64.38 (0.22) | 75.99 (0.04) | 89.62 (0.09) | 50.69 (0.14) | **36.40** (0.13)* |
| | Ours(Para) | 64.14 (0.27) | 75.85 (0.08) | 89.53 (0.07) | 51.24 (0.09)* | 36.16 (0.19)* |
| Llama3-Instruct$_{70B}$ | LN-PE | 71.30 (0.17) | 87.42 (0.09) | 89.30 (0.15) | 67.32 (0.17) | 43.98 (0.27) |
| | INSIDE | 68.40 (0.32) | 83.83 (0.16) | 88.32 (0.10) | 65.19 (0.30) | 42.96 (0.41) |
| | M.I. | 72.75 (0.22) | 88.26 (0.10) | 90.28 (0.15) | 67.11 (0.20) | 43.91 (0.37) |
| | Semantic Entropy | **73.75** (0.50) | 88.77 (0.12) | 89.44 (0.11) | 67.83 (0.20) | 44.33 (0.23) |
| | Semantic Density | 72.56 (0.44) | 86.93 (0.31) | 88.50 (0.39) | 64.72 (0.72) | 42.65 (0.38) |
| | SAR | 73.26 (0.29) | 88.79 (0.12) | 90.67 (0.09) | 68.29 (0.13) | 44.15 (0.41) |
| | Ours(SOC) | 72.95 (0.32) | 89.09 (0.05)* | 91.04 (0.05)* | 69.03 (0.15)* | **44.98** (0.34)* |
| | Ours(Para) | 72.34 (0.32) | **89.41** (0.05)* | **91.07** (0.08)* | 69.31 (0.12)* | 44.81 (0.19)* |

Table 12: Generation correctness prediction results evaluated with Selective Accuracy at $90\%$ Coverage (Acc@90), where a larger value indicates better UQ performance. Each method is evaluated 10 times on each dataset for each base model. The score outside the brackets represents the mean of the 10 trials, while the score inside the brackets indicates the standard deviation. The **bold** number represents the best performance across all methods. The underline highlights the mean value that outperforms all baselines. The asterisk $*$ indicates that the number is statistically significantly better than the SOTA baseline (SAR) at the $5\%$ significance level. The numbers in the 'Avg. Improvement' row show the average mean improvement and reduction in standard deviation compared to the SOTA baseline across models.

| Models | UQ methods | SciQ Acc@90 | TriviaQ Acc@90 | CoQA Acc@90 | AmbigQA Acc@90 | TruthfulQA Acc@90 |
|---|---|---|---|---|---|---|
| Llama2-chat$_{7B}$ | LN-PE | 51.66 (0.18) | 61.48 (0.07) | 80.06 (0.12) | 43.93 (0.09) | 42.56 (0.18) |
| | INSIDE | 50.42 (0.23) | 61.03 (0.07) | 79.85 (0.07) | 43.04 (0.16) | 40.41 (0.26) |
| | M.I. | 50.83 (0.20) | 61.69 (0.05) | 80.46 (0.05) | 43.69 (0.14) | 42.12 (0.30) |
| | Semantic Entropy | 52.01 (0.21) | 61.78 (0.07) | 80.82 (0.07) | 43.84 (0.17) | 39.65 (0.25) |
| | Semantic Density | 51.54 (0.26) | 61.46 (0.11) | 80.43 (0.24) | 43.36 (0.20) | 39.62 (0.27) |
| | SAR | 51.82 (0.35) | 61.66 (0.08) | 80.89 (0.15) | 43.75 (0.17) | 42.29 (0.45) |
| | Ours(SOC) | **52.41** (0.12)* | 61.99 (0.05)* | **81.17** (0.07)* | 44.29 (0.08)* | 43.77 (0.11)* |
| | Ours(Para) | 52.38 (0.11)* | **62.26** (0.02)* | 81.14 (0.07)* | **44.66** (0.07)* | **43.80** (0.10)* |
| Mistral-Instruct$_{12B}$ | LN-PE | 61.62 (0.24) | 73.00 (0.05) | 89.14 (0.11) | 48.65 (0.15) | 42.00 (0.33) |
| | INSIDE | 60.97 (0.29) | 72.78 (0.07) | 88.78 (0.10) | 47.48 (0.14) | 41.31 (0.19) |
| | M.I. | 60.93 (0.22) | 73.17 (0.04) | 89.31 (0.08) | **48.84** (0.14) | 41.67 (0.28) |
| | Semantic Entropy | **62.10** (0.21) | **73.49** (0.04) | **89.99** (0.05) | 48.43 (0.10) | 42.15 (0.14) |
| | Semantic Density | 61.20 (0.27) | 72.56 (0.07) | 88.88 (0.26) | 48.09 (0.28) | 41.51 (0.34) |
| | SAR | 61.79 (0.19) | 73.09 (0.06) | 89.96 (0.14) | 48.62 (0.16) | **42.37** (0.20) |
| | Ours(SOC) | 61.72 (0.19) | 72.91 (0.06) | 89.48 (0.08) | 48.55 (0.07) | 41.96 (0.23) |
| | Ours(Para) | 61.98 (0.16)* | 73.00 (0.04) | 89.38 (0.07) | 48.83 (0.09)* | 41.82 (0.15) |
| Llama3-Instruct$_{8B}$ | LN-PE | 59.82 (0.40) | 70.30 (0.03) | 86.22 (0.14) | 46.71 (0.13) | 33.61 (0.26) |
| | INSIDE | 58.96 (0.27) | 69.34 (0.12) | 85.82 (0.06) | 46.02 (0.14) | 32.19 (0.27) |
| | M.I. | 60.21 (0.17) | 70.26 (0.06) | 86.42 (0.09) | 46.65 (0.11) | 33.32 (0.12) |
| | Semantic Entropy | **60.82** (0.21) | **70.70** (0.07) | 86.84 (0.08) | 46.71 (0.15) | 33.46 (0.27) |
| | Semantic Density | 60.07 (0.30) | 69.97 (0.13) | 85.57 (0.49) | 46.30 (0.31) | 33.39 (0.35) |
| | SAR | 60.41 (0.22) | 70.49 (0.05) | **87.30** (0.14) | 46.81 (0.14) | 33.66 (0.25) |
| | Ours(SOC) | 60.36 (0.18) | 70.56 (0.04)* | 87.11 (0.06) | 46.94 (0.08)* | **33.69** (0.13) |
| | Ours(Para) | 60.79 (0.22) | 70.53 (0.05)* | 87.09 (0.04) | **47.19** (0.08)* | 33.46 (0.08) |
| Llama3-Instruct$_{70B}$ | LN-PE | 69.02 (0.35) | 85.25 (0.11) | 88.15 (0.09) | 64.75 (0.15) | 41.63 (0.24) |
| | INSIDE | 66.92 (0.39) | 83.71 (0.09) | 87.88 (0.09) | 63.58 (0.24) | 41.02 (0.22) |
| | M.I. | 68.92 (0.20) | 85.16 (0.04) | 89.12 (0.08) | 63.70 (0.16) | 41.58 (0.28) |
| | Semantic Entropy | **70.29** (0.18) | **86.30** (0.07) | **89.44** (0.05) | 64.78 (0.11) | 41.59 (0.14) |
| | Semantic Density | 69.81 (0.23) | 84.01 (0.29) | 87.35 (0.10) | 63.75 (0.28) | 41.25 (0.32) |
| | SAR | 70.22 (0.23) | 85.98 (0.06) | 89.16 (0.10) | 64.67 (0.17) | 41.50 (0.23) |
| | Ours(SOC) | 70.19 (0.11) | 86.10 (0.04)* | 88.96 (0.06) | 65.13 (0.12)* | **41.81** (0.16)* |
| | Ours(Para) | 69.41 (0.21) | 86.13 (0.05)* | 88.96 (0.04) | **65.34** (0.10)* | 41.73 (0.19)* |

Table 13: Average per-example runtime (in seconds) of paraphrasing used in ESI(Para)

| Paraphrase methods | SciQ | TriviaQ | TruthfulQA |
|---|---|---|---|
| DeepSeek API | 12.09 | 12.53 | 11.23 |
| Llama3-Instruct$_{8B}$ | 0.57 | 0.65 | 0.57 |

