# OpenReview forum: "ESI: Epistemic Uncertainty Quantification via Semantic-preserving Intervention for Large Language Models"
_ICLR.cc/2026/Conference — Submitted to ICLR 2026_

### Official Review · Reviewer_1gWk · 2025-10-29

**Soundness:** 2
**Presentation:** 3
**Contribution:** 3
**Rating:** 6
**Confidence:** 5

**Summary:**

I have reviewed this manuscript in a previous conference. I tried to compare the current version and the previous version, but did not find significant changes. I tend to maintain my previous evaluations.

This manuscript presents a new framework that quantifies the epistemic uncertainty from the perspective of invariance to input intervention. The original prompt is intervened in a semantic-preserving manner, then the token-level distribution changes are quantified as an estimation of the epistemic uncertainty in the original answer. The effectiveness of the proposed approach is empirically verified by comparing to several baselines in multiple datasets. An ablation study is conducted to investigate the effect of different choices of algorithmic components.

**Strengths:**

1. The current manuscript is well organized. The motivation, methodology and experimental results are presented in a clear and understandable way.
2. The motivation that “the uncertainty of LLMs can be effectively quantified by evaluating the degree to which the model relies on semantic causal relationships in its inference process.” is interesting and inspiring. It provides a new perspective to help understand where the LLM uncertainty comes from.
3. The usage of Hellinger distance to measure the distribution distance is new. Previous works usually used KL divergence, but this work make a new choice with solid justification, from both theoretical and empirical perspectives.

**Weaknesses:**

1. One major limitation of this work is that when measuring the output distribution changes, it only considers the token-level change, without any analysis of the semantic meaning changes. Given the nature that same semantic meaning can be represented using different words/phrases, what really matters in LLM uncertainty measurement should be the change in semantic space. The current algorithmic design completely ignores this aspect. It would be more compelling if the variance after prompt intervention can be measured in semantic space.
2. The two proposed alternatives for prompt intervention have some inherent limitations. Regarding Skip-One-Char (SOC), skipping several unimportant characters does not really alter the superficial linguistic structures. Instead, it is testing the robustness of the LLM to miss-spelling of the words, which is not the true intention of the framework. Regarding the paraphrasing models, the additional computational cost is indeed expensive, if we want to have a reliable paraphrasing. Experimental results regarding the additional computational cost are missing.
3. The proposed metric is a response-level uncertainty metric, but the compared baselines are mostly entropy-based prompt-level uncertainty metrics, which are usually performing worse than response-level counterparts. SOTA response-level uncertainty metrics are missing from the current experiments, e.g., semantic density [1], Degree [2].

[1] Xin Qiu, Risto Miikkulainen. Semantic density: Uncertainty quantification for large language models through confidence measurement in semantic space, NeurIPS 2024

[2] Zhen Lin, Shubhendu Trivedi, Jimeng Sun, Generating with Confidence: Uncertainty Quantification for Black-box Large Language Models, Transactions on Machine Learning Research, 2024

**Questions:**

1. As mentioned above in point 1 of “weaknesses”, only considering the token-level distribution shift seems to be insufficient for tackling language tasks, in which semantic meaning is clearly more important. Can you add more discussions regarding this point? Do you have any thoughts about how to compensate for this limitation?
2. As discussed in point 2 of “weaknesses”, the SOC is more like testing the robustness of the underlying LLM in tackling miss-spelled or distorted words, instead of superficial linguistic structures alterations. Could you elaborate more on how this potential issue can be mitigated? Can you design some empirical studies to investigate whether the above proposition is true?
3. In addition to the existing prompt-level baselines, more SOTA response-level uncertainty metrics should be included in the experiments to further verify the effectiveness of the proposed framework, e.g., semantic density [1], Degree [2]. Can you add more comparisons?
4. Since the main drawback of the paraphrasing models are their additional computational time, you should include them in the Table 2 as well, e.g., both the weak version and strong version. This will provide the readers with a clearer picture about what is the cost of having a stronger performance, and what are the tradeoffs between computation time and performance gain.

[1] Xin Qiu, Risto Miikkulainen. Semantic density: Uncertainty quantification for large language models through confidence measurement in semantic space, NeurIPS 2024

[2] Zhen Lin, Shubhendu Trivedi, Jimeng Sun, Generating with Confidence: Uncertainty Quantification for Black-box Large Language Models, Transactions on Machine Learning Research, 2024

---

> ### Author Response · Authors · 2025-11-21
> **Author Response I**
>
> We sincerely thank the reviewer for evaluating our work again and for the positive assessment and insightful feedback. As the reviewer noted, the core idea and methodology remain unchanged from the earlier version. In this submission, our revisions focus primarily on strengthening the theoretical justifications in Sections 2 and 3.3. We revised the explanation in Section 2 to improve clarity and logical flow, and reorganized Section 3.3 to make the reasoning more coherent. We also toned down a few claims to avoid potential misunderstandings raised in the previous round of feedback and removed some content to the appendix because of space limitations.
> Below is our response to your comments, which we hope will answer your questions and concerns.
>
> > ### ***1. Response to Question 1***
>
> Thank you for your valuable feedback. We agree that measuring variability in the semantic space of full outputs would, in principle, be ideal for Language Models. However, this is ***computationally prohibitive*** because:
>
>    - The sentence-level output distribution $p(y|x)$ is enormous and cannot be explicitly computed.
>    - Estimating the semantic output distribution $p(S(y)|x)$ requires sampling and embedding-based clustering, which is computationally expensive for modern LLMs.
>    - Estimating the KL divergence between two sampled sequence-level output distributions, $D_{KL}(p(y|x,\tilde{x_1})||p(y|x,\tilde{x_2}))$, suffers from extremely high variance because both distributions are sparse Monte-Carlo reconstructions over an exponential space.
>
> Given these limitations, our choice of token-level distribution is a ***practical trade-off***: it provides full accessibility to the underlying probability distribution, avoids sampling-induced variance, and enables parallel computation across positions, which brings ***better numerical stability and computational efficiency***. Our method can be viewed as repeatedly intervening on the effective “prompt” for each predicted token (i.e., the combination of ​x and $y_{<t}$) and measuring how the model’s local predictive behavior shifts under semantically equivalent inputs.
>
> We agree that developing an estimator that incorporates richer sentence-level semantic structure is an exciting direction for future work. Nonetheless, the current token-level formulation is stable, computationally efficient, and empirically validated, and thus appropriately supports the contributions of our method in this submission.
>
> > ### ***2. Response to Question 2***
>
> Thank you for your valuable feedback. SOC is proposed as an efficient, model-free alternative to paraphrasing for the semantic-preserving intervention function. The only requirement for an intervention function in our framework is that it generates variants that the model interprets as having the same underlying semantics, since no matter how you manipulate the input, as long as the semantics remain, the output should remain invariant for the ground-truth model, then our arguments follow.
>
> Our results in Table 3 empirically show that SOC maintains semantic equivalence to the original inputs at a high rate, indicating that it satisfies the semantic-preservation requirement in practice. Accordingly, SOC is a suitable intervention function for our method.
>
> Therefore, measuring distributional divergence under SOC-generated interventions is fully consistent with the principled motivation of our approach, and the experimental results confirm that SOC-based ESI performs effectively.

---

> ### Author Response · Authors · 2025-11-21
> **Author Response II**
>
> > ### ***3. Response to Question 3***
>
> Thank you for your valuable suggestion. Comparison with Degree (denoted as Self-Con(DegMat)) can be found in Appendix E.1 (Table 8) with several more black-box baselines.
>
> For Semantic Density, we have now finished complete experiments under the same settings as in the main experiment (Table 1). ***We will add it to the main experiments in the revision***. The outcomes are as follows:
>
> |Model| Method               | SciQ             | TriviaQ          | CoQA             | AmbigQA          | TruthfulQA       |
> |-| -------------------- | ---------------- | ---------------- | ---------------- | ---------------- | ---------------- |
> |Llama2-Chat-7B| **Semantic Density** | 70.45 (0.56)     | 76.73 (0.41)     | 72.21 (0.69)     | 69.72 (0.57)     | 56.63 (0.86)     |
> || **SAR**              | 73.40 (0.55)     | 79.45 (0.17)     | 70.72 (0.39)     | 70.59 (0.42)     | 61.11 (0.51)     |
> || **Ours (SOC)**       | **75.30 (0.28)** | 80.22 (0.05) | 71.69 (0.13)     | 72.29 (0.11) | 66.82 (0.18) |
> || **Ours (Para)**      | 75.15 (0.22) | **81.38 (0.07)** | **73.70 (0.09)** | **73.57 (0.12)** | **67.51 (0.22)** |
> |Mistral-Nemo-Instruct| **Semantic Density** | 69.66 (0.88)     | 75.40 (0.70)     | 75.21 (0.90)     | 69.75 (1.08)     | 63.01 (0.72)     |
> || **SAR**              | 77.14 (0.54)     | 84.91 (0.09)     | 79.58 (0.35)     | 75.32 (0.43)     | 68.39 (0.64)     |
> || **Ours (SOC)**       | **77.45 (0.15)** | 83.98 (0.07)     | **79.81 (0.07)** | 75.24 (0.11)     | **71.84 (0.24)** |
> || **Ours (Para)**      | 77.21 (0.20)     | **85.31 (0.04)** | 79.70 (0.14)     | **76.99 (0.11)** | 71.01 (0.24)     |
> |Llama3-Instruct-8B| **Semantic Density** | 69.43 (0.88)     | 77.09 (1.07)     | 70.91 (0.44)     | 69.17 (0.85)     | 63.10 (0.90)     |
> || **SAR**              | 74.73 (0.45)     | 84.39 (0.12)     | 75.05 (0.42)     | 76.58 (0.38)     | 68.02 (0.66)     |
> || **Ours (SOC)**       | **75.24 (0.27)** | 84.77 (0.04) | 77.00 (0.08) | 79.31 (0.16) | 68.59 (0.27) |
> || **Ours (Para)**      | 75.06 (0.21) | **85.19 (0.04)** | **77.66 (0.09)** | **80.37 (0.16)** | **69.58 (0.23)** |
> |Llama3-Instruct-70B| **Semantic Density** | 62.86 (0.67)     | 70.12 (0.61)     | 68.31 (1.07)     | 63.56 (0.64)     | 59.42 (0.96)     |
> || **SAR**              | 68.59 (0.62)     | 76.31 (0.16)     | 68.16 (0.48)     | 68.77 (0.43)     | 65.08 (0.59)     |
> || **Ours (SOC)**       | 70.56 (0.10) | 79.40 (0.04) | 72.98 (0.08) | 72.83 (0.16) | 67.40 (0.14) |
> || **Ours (Para)**      | **71.88 (0.38)** | **80.61 (0.06)** | **75.45 (0.09)** | **74.26 (0.13)** | **69.23 (0.22)** |
>
> As shown above, our proposed estimator still consistently outperforms all baselines in most settings.
>
> > ### ***4. Response to Question 4***
>
> Thank you for your valuable suggestion. Below are the average times (seconds per prompt) used for paraphrasing.
>
> | Para model| SciQ | TriviaQ | TruthfulQA |
> | --------- | ---- | ------- | ---------- |
> | Deepseek API| 12.09 | 12.53 | 11.23|
> | Llama3-Chat-8B| 0.57 | 0.65 | 0.57|
>
> It is worth noting that the latency of API responses is affected by many factors, so there is a lot of room for optimization. The paraphrasing speed of Llama3-Chat-8B can also be optimized (for example, use diverse beam search instead of a long few-shot prompt), but it is beyond the scope of this paper.
>
> We will add this discussion to the revision.
>
> ### ***We hope that our clarifications resolve the reviewer’s concerns. If any aspects of our explanation remain unclear, or if there are further concerns, we would be happy to address them or provide additional experiments. We thank the reviewer again for their time and constructive feedback.***

---

> ### Author Response · Authors · 2025-11-26
> **Summary of Changes in the Revised Submission**
>
> We have revised our original submission based on your valuable feedback and our responses. Specifically:
>
> - We add Semantic Density to the main experiments, including a new introduction sentence in baselines(Lines 303–304), updates to Table 1, and corresponding implementation details in Appendix C.4 (Lines 951–954).
> - We add the experiment of average runtimes (seconds per prompt) used for paraphrasing in Appendix E.4 and Table 13.
>
> We sincerely thank you for your constructive feedback. If there are any remaining concerns, we would be happy to address them and provide further clarification or additional experiments.

---

### Official Review · Reviewer_16Kv · 2025-10-31

**Soundness:** 2
**Presentation:** 3
**Contribution:** 2
**Rating:** 4
**Confidence:** 2

**Summary:**

The paper proposes ESI, a grey‑box uncertainty score for LLMs that probes invariance to semantic‑preserving prompt interventions. Given a prompt $\mathbf{x}$, ESI decodes one answer $\mathbf{y}^\star$, generates paraphrase/character‑edit variants $\tilde{\mathbf{x}}$, and measures the token‑level shift in predictive distributions along the fixed path $\mathbf{y}^\star$ before versus after intervention (Hellinger distance on top‑$k$ logits). The authors argue this approximates an epistemic measure, and show strong AUROC and runtime gains versus Semantic Entropy, SAR, and other baselines across several QA datasets.

**Strengths:**

- Simple, fast, and effective: Teacher‑forced, token‑space distances make ESI markedly cheaper than output‑sampling methods while improving AUROC for error detection.

- Good empirical coverage: Multiple LLM families and both single‑answer and multi‑answer QA datasets.

- Clearly an intuitive method. Using semantic‑preserving input variants to probe model brittleness is conceptually clean.

- Well presented; I didn't know this area before reviewing and felt like I could follow along and learnt some things.

**Weaknesses:**

Due to my lack of familiarity with the area, I mainly focus on the properties of the derived estimator and the evaluations.

- The derivation of the estimator starts from a pairwise divergence over two independent semantic variants, then replaces it with a one‑sided expectation where every pair includes the original prompt as an anchor and distances are computed along the anchor’s decoded path. This changes the sampling measure in input space (all pairs now include the anchor) and output space (the conditioning path shifts to the anchor’s greedy decode). The assumption that the identity variant has non‑zero probability does not remove this bias. As derived, the resulting quantity does not appear to share the same functional as the stated pairwise measure.

- The estimator is derived from KL‑based EPKL, but the algorithm uses Hellinger distance on truncated (top‑k) token distributions with entropy weights; arguments that lean on KL‑style structure do not automatically apply. The object justified in the derivation and the object actually computed are different, so the current version of the paper does not establish properties of the implemented score.

- After the chain rule, expectations over model prefixes are approximated by distances along one decoded answer string, and in the final estimator this is fixed to the anchor prompt’s greedy decode. In high‑entropy or multi‑answer regimes, semantic variants can steer decoding into different modes; measuring only along the anchor path can hide dispersion that would be visible under paths typical for the variants. This makes the score optimistic on ambiguous items and blurs the line between epistemic effects and artifacts of the chosen path.

- Results are reported mainly as AUROC. AUROC alone does not characterize risk–coverage trade‑offs or calibration needed for selective generation and abstention. Without risk–coverage curves, AURC, selective accuracy at fixed coverage, it’s hard to judge how well ESI would perform in real decision pipelines.

**Questions:**

I’m not very familiar with the area and am happy to discuss my review with the authors in detail. The questions below focus on aligning the derivation with the implemented score and on a few evaluation checks.

- Can you align the distance used in the method with the distance used in the derivation (e.g., re‑derive the estimator in the same metric you implement), or justify the mismatch explicitly?

- Could you justify using AUROC alone and, if feasible, add risk–coverage curves, AURC, and selective accuracy at fixed coverage levels?

- Is there a way to quantify the estimator’s bias and variance via a small ablation (e.g., symmetric‑pair comparisons and repeated sampling)?

- Can you compute ESI over a few decoded paths per prompt and aggregate by mean and by max to assess path bias?

---

> ### Author Response · Authors · 2025-11-21
> **Author Response I - Clarification on Concepts**
>
> Thank you for your thoughtful feedback and constructive suggestions. We appreciate the reviewer’s careful evaluation and acknowledgment of our work. Below is our response to your comments, which we hope will answer your questions and concerns.
>
> Before addressing the reviewer’s detailed concerns, we would like to clarify two conceptual points to avoid confusion about notation and the relationship between the derivation and the implemented estimator.
>
> - In the original submission, the symbol $U_{ESI}$ was inadvertently used to refer to two different quantities.
>    - In Section 3.3 and Appendix A, the $U_{ESI}$ refers to the quantity introduced in Section 2 (described from line 189 - line 191 in the original submission), which is the combination of equations 1 & 2:
>       >  $U_{ESI} = E_{\tilde{x}\sim f_{I}}[\frac{1}{N} \sum_{t=1}^{N} D(p(y|y_{<t}, x)||p(y|y_{<t}, \tilde{x}))]$
>
>       When we claim that the ESI estimator approximates the Expected Pairwise KL-Divergence (EPKL), we mean that the above quantity ***with KL-divergence as the distance measure function*** approximates the Expected Pairwise KL-Divergence (EPKL), as explicitly stated in line 803, Appendix A in the original submission. The explicit reference to KL as the distance measure was included in section 3.3 in an earlier draft but was inadvertently removed during the restructuring of Section 3.3; we will restore this clarification in the revision.
>
>   - The estimator used in practice (line 259, section 3.4 in the original submission), which incorporates Hellinger distance, truncated top-k distributions, and entropy weighting, was introduced to provide a ***numerically stable and computationally efficient*** implementation (which is important for practical usage) in large-scale LLM settings. ***We do not claim that this practical variant is a theoretical estimator of EPKL***. To avoid notational confusion, we will denote the practical version by $\tilde{U}_{ESI}$ in the revision.
>
> - The goal of our work is NOT to construct a theoretically perfect estimator of epistemic uncertainty, but to develop ***a practical, computationally efficient, and empirically effective measure*** tailored to the behavior of ***modern LLMs***. The theoretical elements in Section 3.3 are intended as conceptual motivation, rather than strict axiomatic derivations.
>
>    Section 3.3 explains that epistemic uncertainty for LLMs can be approximated by the expected sequence-level distributional shift induced by semantically equivalent variants of the input. Directly estimating this quantity is computationally ***infeasible*** for LLMs. Therefore, we use the expected token-wise distribution shift as a tractable proxy. Although this approximation is biased with respect to the ideal sequence-level divergence, it ***preserves the core intuition and is dramatically more efficient and stable in practice.***
>
>    Moreover, we show that when choosing KL divergence as the underlying distributional distance, the transition from the sequence-level expression to our token-level estimator can be straightforwardly derived, although the transformation is still biased. ***The additional practical modifications (anchoring one prompt–response pair, Hellinger distance, truncated token distributions with entropy weights) do not alter the underlying principle: ESI still estimates the average token-level predictive shift under semantic-preserving input perturbations.***
>
>    For these reasons, we believe our estimator is conceptually clear, computationally feasible, and empirically validated, and that ESI remains a faithful estimator of epistemic uncertainty for LLMs.

---

> ### Author Response · Authors · 2025-11-21
> **Author Response II -Response to Weakness 1**
>
> > ### ***1. Response to Concerns about the derivation in Appendix A, which claims that ESI estimator with KL-divergence as the distance measure function approximates the Expected Pairwise KL-Divergence (first point in Weakness)***
>
> Yes, the approximation is biased. As stated above, the theoretical justification in Section 3.3 is not to claim a perfect unbiased estimator of the epistemic uncertainty of LLMs (which we believe is infeasible at the current stage), but to justify that our proposed practical estimator based on expected token-wise distribution shift indeed measures epistemic uncertainty. And the derivation in the Appendix provides a justification that our proposed token-wise estimator (with KL-divergence as distribution distance measure) is indeed a biased but reasonable approximation of a more ideal but impractical estimator (the expected pair-wise sequence-level distributional shift) of epistemic uncertainty of LLMs.
>
> To address the reviewer’s concern, we clarify the two key approximation steps in the derivation and why both are reasonable:
>
> - ***Approximating sequence-level divergence using average token-wise divergence***
>
>   > $\sum_{t} D_{KL}(p(y|y_{<t}, \tilde{x_1})||p(y|y_{<t},\tilde{x_2})) \approx \sum_{t} E_{y_{<t}}[D_{KL}(p(y|y_{<t}, \tilde{x_1})||p(y|y_{<t},\tilde{x_2}))] = D_{KL}(p(y|\tilde{x_1})||p(y|\tilde{x_2}))$
>
>   , where y is a response sampled from $\tilde{x_1}$.
>
>   Since computing the expectation in the middle term is intractable, we follow the common practice used in equation 9 of [1] and approximate it using a single realization $y \sim p(\cdot|\tilde{x_1})$.
>
>   This introduces bias, but it is a standard and practical approximation that preserves the structure of the underlying KL decomposition.
>
>   We also note that the original submission mentioned using the greedy response of $\tilde{x_1}$ as the single realization (Appendix A line 790 in the original submission), but this is not necessary. As stated in the main text, our estimator does not require greedy decoding (although non-greedy may introduce additional bias and variance). We will revise this in the revision.
>
> - ***Introducing the anchor variant, which fixes $\tilde{x_1} = x'$, the particular prompt for which uncertainty is being evaluated, and only samples $\tilde{x_2}$.***
>
>   We agree that this changes the sampling measure and introduces bias. However, it remains a reasonable approximation:
>
>   Since $x'$ is trivially semantically equivalent to itself, and $y'$ is a response generated from $x'$. Thus, anchoring corresponds to conditioning the semantic-variant distribution $f_{I}$ on the specific instance $x'$, which is precisely the scenario in which epistemic uncertainty is evaluated.
>
>   Because we do not make any assumption on the distribution over semantic-preserving variants $f_{I}$. The anchoring can be considered as sampling from a posterior distribution given the actual prompt $x'$. This avoids unnecessary variance, prevents degenerate samples, and offers substantial numerical advantages.
>
>   Similarly, if we care about the uncertainty of a certain prompt–response pair ($x'$, $y'$ ) (like the situation in the experiments), anchoring on both $x'$ and $y'$ is a natural choice which offers numerical advantages (e.g., the computational efficiency through teacher forcing).
>
> These two approximations convert an intractable sequence-level divergence into a tractable token-level estimator that retains the intended functional meaning: measuring predictive instability under semantic-preserving perturbations of the input. Despite the bias, the estimator remains principled and is strongly validated by empirical performance.
>
> [1] Malinin et al. (2021). Uncertainty Estimation in Autoregressive Structured Prediction. ICLR 2021.

---

> ### Author Response · Authors · 2025-11-21
> **Author Response III - Response to Weakness 2&3 and Question 1**
>
> > ### ***2. Response to Concerns about alignment between the derivation and the implemented score  (second point in Weakness and the first point in questions)***
>
> We appreciate the reviewer’s observation and agree that the practical estimator in section 3.4 is not identical to the KL-based theoretical quantity used in the derivation. We clarify below how the two are related and why the theoretical development still justifies the implemented score.
>
> ***KL-based derivation motivates the structure of the estimator, not its exact functional form***. Section 3.3 explains that epistemic uncertainty for LLMs can be approximated by the expected sequence-level distributional shift induced by semantically equivalent variants of the input. Derivation in the appendix showed that, when KL is used as the distance measure, the expected sequence-level predicted divergence can be approximated by expected token-wise predictive divergences via the chain rule of KL.
>
> This establishes the structural foundation, namely, that ***epistemic uncertainty can be estimated through average token-wise distribution shifts across semantic variants***. It is this structural insight, not the specific choice of distribution distance measure function, that the practical estimator inherits. As shown in section 3.4, the practical estimator measures the importance-weighted token-level predictive shifts across semantic variants.
>
> Given the practical purpose of our work, the practical implementations are designed to ensure computational stability and efficiency, not to change the underlying principle. The empirical results validate the effectiveness of the implemented score.
>
> > ### ***3. Response to Concerns about the anchoring decoding path (third point in Weakness)***
>
> We thank the reviewer for the insightful feedback. We agree that anchoring the computation to a single decoded path introduces bias, as stated above. In principle, one could sample multiple decoded paths and compute ESI along each path, then aggregate the results. This may provide a more complete picture of distributional dispersion.
>
> However, this procedure is ***extremely expensive*** for modern LLMs: sampling M trajectories requires additional decoding runs, and computing ESI for each trajectory multiplies the total computation of forward passes by M.
>
> In contrast, ***computational efficiency is one of the main contributions of our proposed method***. Our anchored estimator evaluates epistemic uncertainty along the realized generation path of the model’s actual prediction, which is often the most relevant quantity for instance-level decision making. It also enables teacher forcing, yielding substantial computational savings and improved numerical stability.
>
> Importantly, despite the bias introduced by anchoring, our empirical results show that the implemented estimator is highly effective: it consistently outperforms existing UQ baselines across most settings. This indicates that anchoring does not obscure meaningful epistemic effects in practice.
>
> While exploring multi-path aggregation is an interesting direction for future work, the present anchoring strategy is a principled, efficient, and empirically validated approximation that supports the contribution of our method.

---

> ### Author Response · Authors · 2025-11-21
> **Author Response VI - Response to Question 3&4**
>
> > ### ***4. Response to additional experiments on bias (third and fourth point in questions)***
>
> Thank you for the suggestion. We interpret the reviewer’s question as asking for an empirical estimate of the bias introduced by anchoring—specifically, comparing our anchored estimator (ESI) to a symmetric pairwise sampling procedure that draws two semantic variants independently and samples a response from one of them (i.e., sample $x_1$ and $x_2$, and then sample $y$ from $x_1$). We conducted the following experiments:
>
> 1. Generating semantic variants and multiple decoded paths. For each query, we first generate 5 paraphrases (same as ESI(Para) did), denoted as $x_i$. For each $x_i$, we sample 5 responses $y_{i,j}$ (with top-p=0.9), yielding 25 ($x_i$,  $y_{i,j}$) pairs for each query.
> 2. Computing symmetric pairwise divergences. For every ($x_i$,  $y_{i,j}$) and every other paraphrase $x_k$ with $k!=i$, we compute the pair-wise average token-wise distribution shift as in ESI. We denote the score as $d_{i,j,k}$. For each paraphrase $x_i$, there are 4*5 scores in total. Then, sampling one score from the 4*5 score pool resembles the process of sampling $y$ from $x_i$, sampling $x_k$, and calculating the pair-wise divergence.
> 3. Approximating the pair-wise estimator. For each query, we sample one such score for each paraphrase $x_i$, average them, and treat this as one instance of the symmetric pairwise sampling estimator. We repeat this process 10 times to obtain multiple estimates per query, then average them to get a more robust estimation.
> 4. Measuring bias and variance. For each query, we then compute the difference between the estimated score with the original ESI score as bias. We then calculate the average bias and the variance of the bias for a dataset.
>
> We experiment using LLama3-8b-instruct on TruthfulQA. The results are as follows:
>
> - performance of ESI (AUROC): 70.08
> - performance of symmetric sampling (AUROC): 69.82
> - average value of ESI score: 0.25450
> - average value of symmetric sampling score: 0.26136
> - average bias: 0.00686
> - variance of bias: 0.028965
>
> We hope our interpretation of the reviewer’s question is correct and that the above experiments address the concerns. If additional clarification or further experiments would be helpful, we would be happy to provide them.

---

> ### Author Response · Authors · 2025-11-21
> **Author Response V -More Evaluation Metrics (Weakness 4 and Question 2)**
>
> > ### ***5. Response to Concerns about Insufficient Evaluation Metrics  (fourth point in Weakness and the second point in questions)***
>
> Thanks for the valuable feedback. Our original submission used AUROC primarily due to the space limitation (Table 1 is already very large) and because AUROC is the main or sole metric adopted by most relevant baselines [2, 3, 4, 5]. We fully agree that including additional evaluation metrics can strengthen the robustness of our work.
>
> Following the reviewer’s suggestion, we have added experiments using several commonly used metrics, including AUCPR (Area Under the Precision–Recall Curve), AURC (Area Under the Risk–Coverage Curve), and SA@X (Selective Accuracy at X% Coverage).
> We conduct these extended evaluations on two representative datasets (SciQ and TruthfulQA) and two widely used LLMs (Llama2-Chat-7B and Llama3-8B-Instruct). The results are shown below:
>
> | Model | Method | SciQ AUROC | SciQ AUCPR | SciQ AURC | SciQ SA@90 | SciQ SA@80 | TruthfulQA AUROC | TruthfulQA AUCPR | TruthfulQA AURC | TruthfulQA SA@90 | TruthfulQA SA@80 |
> |-------|---------|-------------|-------------|-------------|------------------|------------------|------------|------------|------------|-----------------|-----------------|
> | **Llama2-Chat-7B** | LN-PE | 71.75 (0.39) | 72.12 (0.81) | 36.06 (0.46) | 51.66 (0.18) | 55.11 (0.48) | 61.78 (0.52) | 71.76 (0.75) | 52.69 (0.42) | 42.56 (0.18) | 44.46 (0.55) |
> | | INSIDE | 61.78 (0.59) | 63.30 (0.50) | 45.84 (0.87) | 50.42 (0.23) | 53.39 (0.21) | 53.99 (0.94) | 63.91 (0.57) | 55.21 (0.92) | 40.41 (0.26) | 40.03 (0.40) |
> | | Semantic Entropy | 71.37 (0.59) | 73.20 (0.64) | 37.74 (1.06) | 52.01 (0.21) | 55.56 (0.32) | 56.24 (0.69) | 61.78 (0.43) | 54.46 (1.48) | 39.62 (0.27) | 40.47 (0.23) |
> | | SAR | 73.06 (0.52) | 73.17 (0.64) | 34.57 (0.46) | 51.82 (0.35) | 55.11 (0.39) | 60.83 (0.57) | 70.33 (0.56) | 52.59 (0.50) | 42.29 (0.45) | 43.45 (0.42) |
> | | **Ours (SOC)** | **75.15 (0.14)** | 75.45 (0.24) | **32.52 (0.16)** | **52.41 (0.12)** | 56.53 (0.24) | 66.80 (0.33) | 77.28 (0.38) | 50.24 (0.18) | 43.77 (0.11) | 46.28 (0.29) |
> | | **Ours (Para)** | 75.10 (0.20) | **76.30 (0.29)** | 32.99 (0.22) | 52.38 (0.11) | **56.95 (0.15)** | **67.22 (0.40)** | **78.00 (0.28)** | **49.42 (0.37)** | **43.80 (0.10)** | **47.47 (0.26)** |
> | **Llama3-8B-Instruct** | LN-PE | 71.41 (0.60) | 65.11 (0.87) | 28.65 (0.54) | 59.82 (0.40) | 63.15 (0.17) | 65.84 (0.76) | 80.98 (0.61) | 58.23 (0.45) | 33.61 (0.26) | 35.90 (0.16) |
> | | INSIDE | 60.56 (0.37) | 55.70 (0.61) | 35.74 (0.48) | 58.96 (0.27) | 60.44 (0.37) | 61.59 (0.54) | 75.67 (0.36) | 59.33 (0.71) | 32.19 (0.27) | 33.45 (0.31) |
> | | Semantic Entropy | 72.02 (0.64) | 69.33 (0.59) | 30.22 (0.94) | **60.82** (0.21) | **64.69 (0.34)** | 66.68 (0.80) | 80.93 (0.62) | 57.98 (1.37) | 33.46 (0.27) | 35.80 (0.34) |
> | | SAR | 74.38 (0.45) | 68.90 (0.80) | 26.70 (0.35) | 60.41 (0.22) | 64.34 (0.53) | 68.00 (0.50) | 82.01 (0.63) | 57.05 (0.55) | 33.66 (0.25) | 35.68 (0.41) |
> | | **Ours (SOC)** | **75.28 (0.27)** | **69.82 (0.20)** | **25.22 (0.16)** | 60.36 (0.18) | 64.38 (0.22) | 68.54 (0.26) | 82.25 (0.20) | 55.70 (0.20) | **33.69 (0.13)** | **36.40 (0.13)** |
> | | **Ours (Para)** | 75.02 (0.22) | 69.60 (0.36) | 25.30 (0.16) | 60.79 (0.22) | 64.14 (0.27) | **69.62 (0.24)** | **82.30 (0.19)** | **54.71 (0.28)** | 33.46 (0.08) | 36.16 (0.17) |
>
> Similar to the main experiments, each method is evaluated 10 times on each dataset for each base model. The score outside the brackets represents the mean of the 10 trials, while the score inside the brackets indicates the standard deviation. The bold number represents the best performance across all methods.
>
> Our methods still consistently outperform all baselines across most settings. We appreciate the reviewer’s suggestion, and we will include these additional evaluations in the appendix to further enhance the robustness and completeness of our empirical study.
>
> [2] Duan et al. (2024). Shifting attention to relevance: Towards the predictive uncertainty quantification of free-form large language models. ACL 2024.
>
> [3] Kuhn et al. (2023). Semantic uncertainty: linguistic invariances for uncertainty estimation in natural language generation. ICLR.
>
> [4] Chen et al. (2024). INSIDE: llms’ internal states retain the power of hallucination detection. ICLR 2024.
>
> [5] Qiu et al. (2024). Semantic density: Uncertainty quantification for large language models through confidence measurement in semantic space. NeurIPS 2024.
>
> ### ***We hope that our clarifications resolve the reviewer’s concerns. If any aspects of our explanation remain unclear, or if there are further concerns, we would be happy to address them or provide additional experiments. We thank the reviewer again for their time and constructive feedback.***

---

> ### Author Response · Authors · 2025-11-26
> **Summary of Changes in the Revised Submission**
>
> We have revised our original submission based on your valuable feedback and our responses. Specifically:
>
> - We add an explicit formula (equation 3 in line 195) of $U_{ESI}$ and change the symbol used for the practical implementation to $U^{*}_{ESI}$ to clearly distinguish them.
> - We revised the description in Section 3.3 (Lines 239 – 245) to explicitly clarify that the theoretical justification in Appendix B is derived under KL divergence as the distance measure. We also moderate the original claims to avoid any potential misunderstanding that we provide an unbiased estimator of epistemic uncertainty.
> - We revised the theoretical justification in Appendix B to clarify the rationale behind the two approximation steps and to explicitly acknowledge the presence of potential bias.
> - We add a paragraph discussing potential estimation errors in Appendix A Limitation (line 795 - 800).
> - We include additional experimental results using complementary evaluation metrics (AURC, AUCPR, selective accuracy at coverage 80%/90%) in Appendix E.3 COMPLEMENTARY EXPERIMENTS ON MORE EVALUATION METRICS.
>
> We sincerely thank you for your constructive feedback. If there are any remaining concerns, we would be happy to address them and provide further clarification or additional experiments.

---

### Official Review · Reviewer_SfBQ · 2025-11-05

**Soundness:** 1
**Presentation:** 2
**Contribution:** 1
**Rating:** 2
**Confidence:** 4

**Summary:**

The authors present a measure of the variability in probabilistic predictions from a large language model (LLM) across augmentations of the LLM’s input. They evaluate how well this measure predicts the correctness of text generated by the LLM on question-answering datasets.

**Strengths:**

Originality: the proposed predictive-variability measure appears to be a new basis for predicting the correctness of LLM generations.

Quality: the authors compare to a good number of baseline methods.

Clarity: the paper is easy to follow at a low level.

Significance: predicting the correctness of model outputs warrants research attention.

**Weaknesses:**

I believe the technical core of this work is unsound.

- First, subjective model-based uncertainties alone are not a legitimate basis for assessing model reliability (Bickford Smith et al, 2025). Predictive uncertainty can be used as an estimator of predictive loss, but this does not require decomposing uncertainty.
- Second, the authors nevertheless appeal to an uncertainty decomposition that has been shown to be incoherent (Bickford Smith et al, 2025), and then reinvent the definition of epistemic uncertainty based on a questionable causal argument and a lot of arbitrary algorithmic decisions.
- Third, even if we forget all of this and just take the proposed measure of predictive variability at face value, it’s unclear how the variability measure reflects the paper’s emphasis on semantics as opposed to tokens. The measure is an average, over input augmentations and token indices, of an entropy-weighted distance between two token distributions. Since the same semantic content can be conveyed as many different token sequences, a fact that motivated key prior work (Farquhar et al, 2024; Kuhn et al, 2023), this is a conceptually flawed measure of predictive variability in semantic space, which is what we care about.

Now, maybe all that really matters is what works in practice. But I’m also unconvinced by the empirical evidence at hand. The authors are trying to solve a binary-classification problem: given an input, is the LLM’s generation correct? The use of AUROC to measure performance in this context has serious issues (Dyrland et al, 2022; Ferrer, 2025; Hand, 2009; Hand & Anagnostopoulos, 2021), so I cannot judge the practical use of the proposed method based on the numerical results provided.

---

Bickford Smith et al (2025). Rethinking aleatoric and epistemic uncertainty. ICML.

Dyrland et al (2022). Does the evaluation stand up to evaluation? A first-principle approach to the evaluation of classifiers. arXiv.

Farquhar et al (2024). Detecting hallucinations in large language models using semantic entropy. Nature.

Ferrer (2025). No need for ad-hoc substitutes: the expected cost is a principled all-purpose classification metric. TMLR.

Hand (2009). Measuring classifier performance: a coherent alternative to the area under the ROC curve. Machine Learning

Hand & Anagnostopoulos (2021). Notes on the H-measure of classifier performance. arXiv.

Kuhn et al (2023). Semantic uncertainty: linguistic invariances for uncertainty estimation in natural language generation. ICLR.

**Questions:**

How does your method compare to the baseline methods if you use performance metrics other than AUROC?

---

> ### Author Response · Authors · 2025-11-18
> **Author Response I - Response to Concerns on Soundness of Technical Core I**
>
> Thank you for your thoughtful feedback and constructive suggestions. We appreciate the reviewer’s careful evaluation of our work. Below we address the comments in detail and hope our responses clarify the questions and concerns raised.
>
> > ### ***1. Response to Soundness of Technical Core***
>
> Before addressing the reviewer’s detailed concerns, we would like to clarify the goal and scope of our work. Our primary objective is ***NOT*** to propose a theoretically perfect estimator of epistemic uncertainty for a general model. Instead, our contribution is to develop a ***practical***, ***computationally efficient***, and ***empirically effective*** operationalization of epistemic uncertainty tailored to the characteristics of ***large language models***. The theoretical components we introduce serve as conceptual motivation rather than strict axiomatic derivations, and the estimator is intentionally designed to be scalable, robust, and useful in real-world LLM applications. With this framing, we address the reviewer’s specific comments below.
>
>
> > ### ***1.1 Response to "reinvent the definition of epistemic uncertainty based on a questionable causal argument"***
>
>   We respectfully disagree with the reviewer’s comment that we “reinvent” the definition of epistemic uncertainty or rely on a problematic causal argument.
>
>   Our work explicitly starts from a well-accepted definition of epistemic uncertainty:
>
>   > ***epistemic (aka systematic) uncertainty refers to uncertainty caused by a lack of knowledge (about the best model)*** [1, 2]
>
>   A standard formulation of this notion is the KL divergence between the learned model $p(y|x)$ and the best or ground-truth model $p^*(y|x)$ (language generation process for a language model):
>
>   > $D_{KL}(p(y|x)||p^*(y|x))$
>
>   which is a common practice for estimating epistemic uncertainty [2, 8]. Hence, our starting point is fully aligned with established definitions.
>
>   However, we argue that this single-input formulation is insufficient for large language models based on our causal argument. Our reasoning follows from a simple and intuitive assumption about human language:
>   > The generated response is causally determined by the semantics of the context text
>
>   Then, based on our causal argument, we have the proposition that
>
>   > semantically equivalent variations of the prompt should not lead to different answer distributions if a model learnt the correct response distributions.
>
>   I respectfully disagree that the causal argument is problematic since the assumption is intuitive and the invariance of causal mechanism/intervention is a well-established theory.
>
>   Thus, if a model outputs the correct distribution for one input $x$, but fails to do so for many of its semantically equivalent variants $\tilde{x}$, then evaluating distribution shift only at $x$ can be misleading. In such cases, $D_{KL}(p(y|x)||p^*(y|x))=0$, but the learned model does not truly capture the ground-truth model, i.e., should have non-zero epistemic uncertainty..
>
>   Therefore, we claim that $E_{\tilde{x}}[D_{KL}(p(y|x,\tilde{x})||p^*(y|x))]$ is a better practice of epistemic uncertainty ***for Language Models***.
>
>   This formulation is NOT a redefinition of epistemic uncertainty, but a ***refinement that operationalizes the standard definition in a way that is better aligned with the characteristics of language models***, namely, the existence of broad semantic invariances in natural language expression.
>
> [1] Hullermeier & Waegeman (2021). Aleatoric and epistemic uncertainty in machine learning: an introduction to concepts and methods. Machine Learning.
>
> [2] Gruber et al. (2023). Sources of uncertainty in machine learning—a statistician’s view. arXiv.
>
> [8] Abbasi-Yadkori et al. (2024). To believe or not to believe your LLM: iterative prompting for estimating epistemic uncertainty. NeurIPS 2024.

---

> ### Author Response · Authors · 2025-11-18
> **Author Response II - Response to Concerns on Soundness of Technical Core II**
>
> > ### ***1.2 Response to "a lot of arbitrary algorithmic decisions"***
>
>   Thank you for your valuable feedback. We respectfully disagree with the reviewer’s comment that our algorithmic decisions are arbitrary. Each decision choice serves the same goal: to transform this theoretically well-motivated but ***intractable*** quantity $E_{\tilde{x}}[D_{KL}(p(y|x,\tilde{x})||p^*(y|x))]$ into a ***practical and efficient*** estimator for ***modern LLMs***. Although this estimator might be imperfect, it is demonstrably more effective and computationally efficient than existing UQ methods for LLMs, as supported by our extensive experiments. Below, we clarify the rationale behind each decision.
>
>   (1) First, since the ground-truth model $p^*(y|x)$ is unknown, inspired by [3], we use EPKL, $E_{\tilde{x_1}, \tilde{x_2}}[D_{KL}(p(y|x,\tilde{x_1})||p(y|x,\tilde{x_2}))]$, to replace the intractable quantity. This measures the expected distributional shift induced by semantically equivalent prompt variants and provides a reasonable surrogate for the original objective.
>
>   (2) Why token-level distributions instead of sequence-level distributions?
>
>   The full sentence-level output distribution $p(y|x)$ for modern LLMs is ***infeasible*** due to the enormous size of the output space. Consequently, measuring distributional shifts at the sequence level would require reconstructing this distribution through extensive sampling, which is both computationally expensive and highly noisy, leading to large variance.
>
>   Instead, our method works on the ***token-level distribution***. This choice brings at least two advantages
>   - ***Numerical Stability***: Compared to the intractable sequence distribution over the entire output space, the token distribution is fully accessible. Therefore, we do not need to approximate it through sampling. In addition, it is more responsive to prompt intervention, as it contains information throughout the vocabulary space.
>   - ***Computation Efficiency***: The process can be easily parallelized by teacher forcing, as all tokens are available in advance, thereby making it more computationally efficient compared to other sampling methods.
>
>   This choice is also theoretically justified by the KL chain rule (Appendix A):
>
>   > $D_{KL}(p(y|x,\tilde{x_1})||p(y|x,\tilde{x_2})) = \sum_{t} E_{y<t}[D_{KL}(p(y|y_{<t}, \tilde{x_1})||p(y|y_{<t},\tilde{x_2}))]$
>
>   Our token-level estimator can therefore be viewed as a biased but principled and computationally efficient approximation to the right-hand side.
>
>   (3) Intervention functions & Distance Measuring functions.
>   The intervention function specifies how semantically equivalent variants $\tilde{x}$ are generated, and the distance metric determines how we quantify divergence between the distributions. All of them are just specific implementations of our proposed methods, which follow the underlying principle: ESI still estimates the average token-level predictive shift under semantic-preserving input perturbations.
>
> We evaluate several principled choices for both components. Para + Hellinger distance is the best-performing configuration based on empirical results rather than arbitrary preference.
>
>   Comprehensive ablations are provided in Figure 4(a) for intervention functions and Appendix D.1 / Table 6 for distance metrics. Across all configurations, our method generally outperforms existing SOTA baselines, demonstrating that the approach is robust to the choice of intervention mechanism and divergence measure.
>
>   (4) Entropy-based importance weighting
>
>   Weighting the output tokens is a common practice in language-model uncertainty estimation [4], given the nature of the language modeling (many uninformative tokens), and has been proven effective. Entropy provides a natural and computationally efficient proxy for informativeness and improves estimator stability.
>
> [3] Schweighofer et al. (2023). Introducing an improved information-theoretic measure of predictive uncertainty. M3L & InfoCog Workshops NeurIPS 23.
>
> [4] Duan et al. (2024). Shifting attention to relevance: Towards the predictive uncertainty quantification of free-form large language models. ACL 2024.

---

> ### Author Response · Authors · 2025-11-18
> **Author Response III - Response to Concerns on Soundness of Technical Core III**
>
> > ### ***1.3 Response to "First, subjective model-based uncertainties alone are not a legitimate basis for assessing model reliability (Bickford Smith et al, 2025)" & "the authors nevertheless appeal to an uncertainty decomposition that has been shown to be incoherent (Bickford Smith et al, 2025)"***
>
>   Thank you for your feedback. Since both concerns come from Bickford Smith et al, 2025, please allow me to address them together. I have carefully and thoroughly read this paper, which provides an insightful and in-depth critique of the popular information-theoretic view on decomposition of predictive uncertainty in Bayesian models, i.e.,
>
>   > $H[y| x, D]= E_{p(w | D)}[ H[y | x, w] ] + I[y, w | x, D]$
>
>   If I understand correctly,  the paper shows that the information-theoretical quantity $I[y, w | x, D]$ (noted as the BALD score in Bickford Smith et al, 2025) may be a poor estimator of how well our learned models match reality, and the estimation error can be large. Two main arguments are that the subjective Bayesian model choice need not match reality, and the possible mismatch between training data and the true data-generating process. Therefore, both components in decomposition above might fail to reflect their intended meanings — aleatoric as intrinsic data noise, and epistemic as lack of knowledge of reality model. Therefore, they claimed that the above decomposition is incoherent and subjective model-based uncertainty is not reliable.
>
>   Importantly, this critique does not target the basic and widely accepted definition of epistemic uncertainty - epistemic uncertainty refers to uncertainty caused by a lack of knowledge (about the best model) [1,2]. Rather, it focuses on Bayesian estimators of epistemic uncertainty that rely on the above decomposition, which is inaccurate and incoherent.
>
>   These concerns do not apply to our method for two reasons:
>   1. ***Our setting is not Bayesian, and our estimator is not based on the information-theoretic decomposition***
>
>      Modern LLMs are not trained as Bayesian models; our estimator does not depend on Bayesian model averaging or the information-theoretic decomposition critiqued by Bickford-Smith et al. Instead, our method is grounded directly in the basic definition of epistemic uncertainty and attempts to directly approximate the quantity through the intrinsic nature of human language, i.e., distributional stability to semantically equivalent perturbations. This is conceptually different from the Bayesian setting.
>
>   2. ***A biased estimator can still be highly effective***
>
>      Even when theoretically imperfect, the information-theoretical quantity $I[y, w | x, D]$ has been shown to be practically useful in many works and also acknowledged by Bickford Smith et al, 2025. Similarly, while our estimator is biased with respect to the ideal epistemic KL quantity, our experiments demonstrate that it is more effective and more computationally efficient than existing UQ methods for LLMs.
>
>   Therefore, the conclusions in Bickford-Smith et al. (2025) ***do not contradict the contributions of our work.*** We thank the reviewer for this feedback. In the revised manuscript, we will add a paragraph to the limitations section to discuss the possible estimation error.
>
> > ### ***1.4 Response to "Predictive uncertainty can be used as an estimator of predictive loss, but this does not require decomposing uncertainty."***
>
>   While predictive uncertainty (estimating $H(p(y|x))$ in LLMs) can certainly serve as a proxy for predictive loss, and many prior works (most of our baselines) follow this approach. We argue that ***estimating the epistemic uncertainty in principle provides a better estimator of predictive loss***, since it directly measures the distance between the predicted model and the ground-truth model.
>
>    In contrast, given the high aleatoric nature of human language, even a perfect model that fully captures the ground-truth generative process would still produce high predictive uncertainty because multiple plausible answers may exist for the same input. This makes predictive-uncertainty–only metrics systematically over-pessimistic, even when the model is performing ideally. In practice, the estimation error might be a problem, but our extensive experiments support that our epistemic-focused estimator generally outperforms predictive-uncertainty based methods.
>
> [1] Hullermeier & Waegeman (2021). Aleatoric and epistemic uncertainty in machine learning: an introduction to concepts and methods. Machine Learning.
>
> [2] Gruber et al. (2023). Sources of uncertainty in machine learning—a statistician’s view. arXiv.

---

> ### Author Response · Authors · 2025-11-18
> **Author Response IV - Response to Concerns on Soundness of Technical Core IV**
>
> > ### ***1.5 Response to "Third, even if we forget all of this and just take the proposed measure of predictive variability at face value, it’s unclear how the variability measure reflects the paper’s emphasis on semantics as opposed to tokens. The measure is an average, over input augmentations and token indices, of an entropy-weighted distance between two token distributions. Since the same semantic content can be conveyed as many different token sequences, a fact that motivated key prior work (Farquhar et al, 2024; Kuhn et al, 2023), this is a conceptually flawed measure of predictive variability in semantic space, which is what we care about."***
>
> Thanks for the thoughtful feedback. We respectfully disagree that our measure is “conceptually flawed”; rather, it is a ***principled and tractable approximation aligned with our stated goal***.
>
> 1. Our method focuses on ***semantic equivalence in inputs***, not outputs.
>
>     The reviewer’s concern seems to assume that our measure attempts to quantify variability in semantic output space. This is not our goal. Our method evaluates how stable the model’s predictive distribution is under semantically equivalent variants of the input. We are not claiming that the output-token space itself encodes semantics; we only use token-level distributions as a tractable representation of predictive behavior.
>
> 3. Sequence-level semantic output distributions are theoretically attractive but ***computationally infeasible***
>
>    In principle, we agree that measuring variability in the semantic space of full outputs would be desirable. But for modern LLMs, this is computationally intractable:
>
>    - The sentence-level output distribution $p(y|x)$ is enormous and cannot be explicitly computed, let alone the semantic output distribution $p(S(y)|x)$.
>    - Estimating the semantic output distribution $p(S(y)|x)$ requires sampling and embedding-based clustering as done in prior work such as [5]. This must be repeated for every variant of input, which is extremely computationally expensive for modern LLMs.
>    - Estimating the KL divergence between two sampled sequence-level output distributions, $D_{KL}(p(y|x,\tilde{x_1})||p(y|x,\tilde{x_2}))$, has substantially higher variance than estimating variation-based metrics such as entropy, $H(p(y|x))$. This is because KL estimation requires reconstructing two sparse Monte-Carlo approximations over an exponentially large output space, making it extremely unstable and sensitive to sampling noise.
>
>
> Instead, our method works on the ***token-level distribution***. This choice brings at least two advantages:
> - ***Numerical Stability***: Compared to the intractable sequence distribution over the entire output space, the token distribution is fully accessible. Therefore, we do not need to approximate it through sampling. In addition, it is more responsive to prompt intervention, as it contains information throughout the vocabulary space.
> - ***Computational Efficiency***: The process can be easily parallelized, as all tokens are available in advance, thereby making it more computationally efficient compared to other sampling methods.
>
> This can be interpreted as repeatedly intervening on the effective “prompt” for each predicted token (i.e., the combination of ​x and $y_{<t}$) and measuring how the model’s local predictive behavior shifts under semantically equivalent inputs.
>
> Although this yields a biased estimator of the ideal sequence-level semantic divergence, it is not conceptually flawed: ***it is a well-defined, tractable probe of distributional stability under semantic-preserving input perturbations.***
>
> In practice, our experiments show that our estimator outperforms state-of-the-art uncertainty measures for LLMs, despite its theoretical bias.
>
> [5] Kuhn et al. (2023). Semantic uncertainty: linguistic invariances for uncertainty estimation in natural language generation. ICLR 2023.

---

> ### Author Response · Authors · 2025-11-18
> **Author Response V - Response to Concerns on Empirical Evidence I**
>
> Thanks for the valuable feedback. Our original submission used AUROC primarily due to the space limitation (Table 1 is already very large) and because AUROC is the main or sole metric adopted by most relevant baselines [4, 5, 6, 7]. We fully agree that including additional evaluation metrics can strengthen the robustness of our work.
>
> Following the reviewer’s suggestion, we have added experiments using several commonly used metrics, including AUCPR (Area Under the Precision–Recall Curve), AURC (Area Under the Risk–Coverage Curve), and SA@X (Selective Accuracy at X% Coverage).
> We conduct these extended evaluations on two representative datasets (SciQ and TruthfulQA) and two widely used LLMs (Llama2-Chat-7B and Llama3-8B-Instruct). The results are shown below:
>
> | Model | Method | SciQ AUROC | SciQ AUCPR | SciQ AURC | SciQ SA@90 | SciQ SA@80 | TruthfulQA AUROC | TruthfulQA AUCPR | TruthfulQA AURC | TruthfulQA SA@90 | TruthfulQA SA@80 |
> |-------|---------|-------------|-------------|-------------|------------------|------------------|------------|------------|------------|-----------------|-----------------|
> | **Llama2-Chat-7B** | LN-PE | 71.75 (0.39) | 72.12 (0.81) | 36.06 (0.46) | 51.66 (0.18) | 55.11 (0.48) | 61.78 (0.52) | 71.76 (0.75) | 52.69 (0.42) | 42.56 (0.18) | 44.46 (0.55) |
> | | INSIDE | 61.78 (0.59) | 63.30 (0.50) | 45.84 (0.87) | 50.42 (0.23) | 53.39 (0.21) | 53.99 (0.94) | 63.91 (0.57) | 55.21 (0.92) | 40.41 (0.26) | 40.03 (0.40) |
> | | Semantic Entropy | 71.37 (0.59) | 73.20 (0.64) | 37.74 (1.06) | 52.01 (0.21) | 55.56 (0.32) | 56.24 (0.69) | 61.78 (0.43) | 54.46 (1.48) | 39.62 (0.27) | 40.47 (0.23) |
> | | SAR | 73.06 (0.52) | 73.17 (0.64) | 34.57 (0.46) | 51.82 (0.35) | 55.11 (0.39) | 60.83 (0.57) | 70.33 (0.56) | 52.59 (0.50) | 42.29 (0.45) | 43.45 (0.42) |
> | | **Ours (SOC)** | **75.15 (0.14)** | 75.45 (0.24) | **32.52 (0.16)** | **52.41 (0.12)** | 56.53 (0.24) | 66.80 (0.33) | 77.28 (0.38) | 50.24 (0.18) | 43.77 (0.11) | 46.28 (0.29) |
> | | **Ours (Para)** | 75.10 (0.20) | **76.30 (0.29)** | 32.99 (0.22) | 52.38 (0.11) | **56.95 (0.15)** | **67.22 (0.40)** | **78.00 (0.28)** | **49.42 (0.37)** | **43.80 (0.10)** | **47.47 (0.26)** |
> | **Llama3-8B-Instruct** | LN-PE | 71.41 (0.60) | 65.11 (0.87) | 28.65 (0.54) | 59.82 (0.40) | 63.15 (0.17) | 65.84 (0.76) | 80.98 (0.61) | 58.23 (0.45) | 33.61 (0.26) | 35.90 (0.16) |
> | | INSIDE | 60.56 (0.37) | 55.70 (0.61) | 35.74 (0.48) | 58.96 (0.27) | 60.44 (0.37) | 61.59 (0.54) | 75.67 (0.36) | 59.33 (0.71) | 32.19 (0.27) | 33.45 (0.31) |
> | | Semantic Entropy | 72.02 (0.64) | 69.33 (0.59) | 30.22 (0.94) | **60.82 (0.21)** | **64.69 (0.34)** | 66.68 (0.80) | 80.93 (0.62) | 57.98 (1.37) | 33.46 (0.27) | 35.80 (0.34) |
> | | SAR | 74.38 (0.45) | 68.90 (0.80) | 26.70 (0.35) | 60.41 (0.22) | 64.34 (0.53) | 68.00 (0.50) | 82.01 (0.63) | 57.05 (0.55) | 33.66 (0.25) | 35.68 (0.41) |
> | | **Ours (SOC)** | **75.28 (0.27)** | **69.82 (0.20)** | **25.22 (0.16)** | 60.36 (0.18) | 64.38 (0.22) | 68.54 (0.26) | 82.25 (0.20) | 55.70 (0.20) | **33.69 (0.13)** | **36.40 (0.13)** |
> | | **Ours (Para)** | 75.02 (0.22) | 69.60 (0.36) | 25.30 (0.16) | 60.79 (0.22) | 64.14 (0.27) | **69.62 (0.24)** | **82.30 (0.19)** | **54.71 (0.28)** | 33.46 (0.08) | 36.16 (0.17) |
>
> Our methods still consistently outperform all baselines across most settings. We appreciate the reviewer’s suggestion, and we will include these additional evaluations in the appendix to further enhance the robustness and completeness of our empirical study.
>
> Notably, our main experiments already cover five datasets and four modern LLMs, and we also provide many additional ablations and analyses in the appendix. Together with the new metrics reported here, we believe the experimental evaluation is extensive and sufficiently comprehensive to demonstrate the practical effectiveness of our proposed method.
>
> [4] Duan et al. (2024). Shifting attention to relevance: Towards the predictive uncertainty quantification of free-form large language models. ACL 2024.
>
> [5] Kuhn et al. (2023). Semantic uncertainty: linguistic invariances for uncertainty estimation in natural language generation. ICLR.
>
> [6] Chen et al. (2024). INSIDE: llms’ internal states retain the power of hallucination detection. ICLR 2024.
>
> [7] Qiu et al. (2024). Semantic density: Uncertainty quantification for large language models through confidence measurement in semantic space. NeurIPS 2024
>
> ### ***We hope that our clarifications resolve the reviewer’s concerns. If any aspects of our explanation remain unclear, or if there are further concerns, we would be happy to address them or provide additional experiments. We thank the reviewer again for their time and constructive feedback.***

---

> ### Author Response · Authors · 2025-11-26
> **Summary of Changes in the Revised Submission**
>
> We have revised our original submission based on your valuable feedback and our responses. Specifically:
>
> - We add a paragraph discussing potential estimation errors in Appendix A Limitation (line 795 - 800).
> - We include additional experimental results using complementary evaluation metrics (AURC, AUCPR, selective accuracy at coverage 80%/90%) in Appendix E.3 COMPLEMENTARY EXPERIMENTS ON MORE EVALUATION METRICS.
> - We moderate the original claims in Section 3.3 to avoid any potential misunderstanding that we provide an unbiased estimator of epistemic uncertainty.
>
> We sincerely thank you for your constructive feedback. If there are any remaining concerns, we would be happy to address them and provide further clarification or additional experiments.

---

### Author Response · Authors · 2025-12-03
**Author Summary for Area Chair (Part VI)**

> ### Why we believe the paper meets the ICLR bar

In summary, we believe that once our rebuttal, additional experiments, and clarifications are taken into account, our submission meets the ICLR acceptance bar:

- It tackles an **important and timely problem**: reliable uncertainty estimation for LLMs.
- It provides a new perspective to understand the uncertainty of LLMs and introduces a **novel, conceptually motivated, and practically usable** estimator, which reviewers acknowledge as original and “conceptually clean”.
- It provides **strong and comprehensive empirical evidence** across multiple models, datasets, and metrics, with additional baselines and evaluation criteria added in response to reviewers’ requests.
- The method is **simple, efficient and effective**, increasing potential impact.

Given that the current scores were assigned ***before*** reviewers could consider our rebuttal and additional experiments, we believe this is a strong borderline paper that merits acceptance.

Thank you again for your time and for carefully considering our submission under these challenging circumstances.

Sincerely,
The Authors

---

### Author Response · Authors · 2025-12-03
**Author Summary for Area Chair (Part III)**

> ### Main Concern of Review 16Kv (score 4)

Reviewer 16Kv’s review is overall positive: they describe our method as “simple, fast, and effective,” our motivation as “conceptually clean,” and our experiments as having “good empirical coverage.” The conservative overall score of 4 appears to be driven mainly by (i) the reviewer’s self-reported lack of expertise in this specific area (“due to my lack of familiarity with the area”; “I’m not very familiar with the area”) and (ii) concerns about potential approximaton bias introduced through the approximation steps in the theoretical justification in Appendix B (Weakness 1 and 3) and (iii) the mismatch between the idealized theoretical quantity and the practically implemented estimator after practical optimization (e.g. replacing KL-divergence with Hellinger distance) (Weakness 2).

Reviewer 16Kv explicitly stated that they were “happy to discuss my review with the authors in detail,” indicating openness to updating their view. Unfortunately, the discussion was frozen before they could engage in that discussion. We believe that, had the discussion continued, these concerns could have been further clarified in light of the rebuttal and additional analysis.

- ***Sketch of Our Responses to Approximation Bias*** (Details in Author Response II (1) and Author Response III (3) to Reviewer 16Kv)

   We agree that our estimator is biased relative to an idealized epistemic uncertainty quantity. However, the goal of our work is **not** to construct a theoretically perfect estimator of epistemic uncertainty, but to develop a **tractable, computationally efficient, and empirically effective** measure tailored to the behavior of modern LLMs. The theoretical analysis in Section 3.3 is intended to justify that our practical estimator—based on expected token-wise distribution shift—provides a tractable and reasonable **proxy** for epistemic-style uncertainty, rather than to claim an unbiased estimator of the true epistemic uncertainty of LLMs (which we believe is infeasible at the current stage). The derivation in Appendix B shows that our token-wise estimator (with KL divergence as the distance measure) is a **biased but reasonable approximation** of a more ideal, but impractical, estimator based on expected pairwise sequence-level distributional shifts.

  In the revised version, we revised the theoretical justification in Appendix B to clarify the rationale behind the two approximation steps and to explicitly acknowledge the presence of potential bias. We also add a paragraph discussing potential estimation errors in Appendix A Limitation (lines 795 - 800). We also moderate the original claims in Section 3.3 to avoid any potential misunderstanding that we provide an unbiased estimator of epistemic uncertainty  (Lines 244 – 245).

- ***Sketch of Our Responses to Mismatch between Theoretical Quantity and the Practical Implementation*** (Details in Author Response III (2) to Reviewer 16Kv)

  Justification in Section 3.3 establishes the ***structural*** insight that epistemic uncertainty can be estimated through average token-wise distribution shifts across semantic variants. It is this structural insight, not the specific choice of distribution distance measure function, that the practical estimator inherits. As shown in section 3.4, the practical estimator measures the importance-weighted token-level predictive shifts across semantic variants. Given the practical purpose of our work, we choose Hellinger distance and other practical design choices to ***ensure computational stability and efficiency***, not to change the underlying principle. The empirical results show that the implemented score is effective and aligns with this structural motivation.

> ### Main Concern of Review SfBQ (score 2)

The main weaknesses raised by reviewer SfBQ are grounded in the critique of the information-theoretic decomposition of predictive uncertainty from the Bayesian view by Bickford Smith et al., 2025 [1].

However, this critique does ***not*** directly apply to our setting. Modern LLM and our method is ***not*** Bayesian, and our estimator does ***not*** rely on the information-theoretic decomposition argument that Bickford Smith et al. analyze. We explained this in detail in our Response 1.1-1.4 to reviewer SfBQ.

Moreover, the primary contribution of our work is to propose a ***practical, computationally efficient, and empirically effective*** uncertainty estimator for modern LLMs. The theoretical analysis is intended as conceptual motivation for interpreting our estimator as providing an epistemic-style estimator; we do not claim to provide an unbiased estimator or a perfect decomposition of total uncertainty. Therefore, the strongest criticisms in this review target a theoretical idealization that we do not claim, rather than the core practical contribution and empirical evidence of the paper.

[1] Bickford Smith et al (2025). Rethinking aleatoric and epistemic uncertainty. ICML 2025.

---

### Author Response · Authors · 2025-12-03
**Author Summary for Area Chair (Part II)**

> ### Addressing Common Concerns
- ***Why measure token-level distribution shift instead of sequence-level semantic shift?*** (1gWk: Weakness 1; SfBQ:  Weakness point 3).

  Reviewers question why we focus on token-level distribution shift rather than measuring changes directly in a sequence-level semantic space, given that the same semantics can be expressed with different surface forms.

  - ***Sketch of Our Responses*** (Details in Author Response IV (1.5) to Reviewer SfBQ and Author Response I (1) to Reviewer 1gWK):

    Our primary goal is to develop a ***practical, computationally efficient, and empirically effective*** uncertainty estimator for modern LLMs. Measuring sequence-level semantic output distributions shift is theoretically attractive but ***computationally infeasible.***

    Our choice of token-level distribution shift is a ***practical trade-off*** that offers ***numerical stability*** and ***computational efficiency***. It can be interpreted as repeatedly intervening on the effective “prompt” for each predicted token (i.e., the combination of $​x$ and $y_{< t}$) and measuring how the model’s local predictive behavior shifts under semantically equivalent inputs.

    We agree that designing estimators that incorporate richer sequence-level semantic structure is an exciting direction for future work. However, the current token-level formulation is aligned with our motivation, is stable, efficient, and empirically validated, and we believe it appropriately supports the main contributions of this submission.

- ***Additional Experiments***
  - ***AUROC alone is insufficient as the Evaluation Metric*** (SfBQ: last Weakness and Question; 16Kv: Weakness 4 and Question 2):
    - ***Sketch of Our Responses*** (Details in Author Response V to Reviewer SfBQ and Author Response V (5) to Reviewer 16Kv):

      Our original submission used AUROC primarily due to the space limitation (Table 1 is already very large), and AUROC is the main or sole metric adopted by most relevant baselines. We fully agree that including additional evaluation metrics can strengthen the robustness of our work and ***include additional experimental results using complementary evaluation metrics (AURC, AUCPR, selective accuracy at coverage 80%/90%) in Appendix E.3 COMPLEMENTARY EXPERIMENTS ON MORE EVALUATION METRICS*** in the revised version. These results are consistent with our AUROC findings and further support our conclusions.

  - ***Additional response-level baselines (Semantic Density and Degree)*** (1gWk: Weakness 3 and Question 3):
    - ***Sketch of Our Responses*** (Details in Author Response II (3) to Reviewer 1gWk):

      Comparison with Degree (denoted as Self-Con(DegMat)) can be found in Appendix E.1 (Table 8) in the original submission, with several more black-box baselines. We add Semantic Density to the main experiments in the revision, including a new introduction sentence in baselines (Lines 303–304), updates to Table 1, and corresponding implementation details in Appendix C.4 (Lines 951–954).

---

### Author Response · Authors · 2025-12-03
**Author Summary for Area Chair (Part I)**

Dear Area Chair,

We would like to thank you for taking the additional time to assess our submission under the unusual circumstances following the OpenReview leak and the subsequent rollback of reviews and scores. We are aware of the substantial workload this creates and are grateful for your efforts.

We would nevertheless like to respectfully request a careful reassessment of our submission. Our paper currently has pre-rebuttal scores of ***6, 4, and 2***, but ***none of the reviewers participated in the discussion phase before the rollback***. We believe the score of 2 (Reviewer SfBQ) is largely driven by a theoretical critique (Bickford-Smith et al., 2025 [1], citation 10) that does not apply to our setting, and the score of 4 (Reviewer 16Kv) comes from a generally positive but cautious review (with confidence 2) from a reviewer who explicitly self-reports limited familiarity with this area and explicitly invited further discussion. We respectfully believe that our rebuttal and clarifications would likely have led to a more favorable evaluation.

> ### Summary of our work:

Our work starts from a simple assumption: ***"human responses are primarily causally determined by the semantics of the context text"***. This motivates us to quantify LLM uncertainty by ***"evaluating the degree to which the model relies on semantic causal relationships in its inference process"*** (Reviewer 1gWk acknowledges that this motivation is interesting and inspiring and "provides a new perspective to help understand where the LLM uncertainty comes from"). Building on this, we propose an uncertainty estimator derived from a causal argument: we evaluate ***the invariance of the model’s output under semantic-preserving interventions on the input***. Reviewer 16Kv describes this approach as “conceptually clean” (Strength 3), and Reviewer SfBQ acknowledges that “the proposed measure appears to be a new basis for predicting the correctness of LLM generations” (Originality strength). Detailed analysis please refer to Section 2.

Concretely, we quantify the average token-wise distribution shift before and after semantic-preserving interventions (paraphrasing (Para) and skip-one-character (SOC)), as described in Sections 3.1 and 3.2. In Section 3.3, we provide a theoretical justification that this average token-wise distribution shift estimator provides a tractable and reasonable proxy for epistemic-style uncertainty, rather than total uncertainty.

Our proposed method excels not only in terms of effectiveness but also in computational efficiency (Reviewer 16Kv describes it as “simple, fast, and effective” in Strengths 1). We conduct extensive experiments—-4 models, 5 datasets spanning three dataset types, 11 baselines (including the 5 additional baselines in Table 7)—-with significance testing to demonstrate the effectiveness of our method, and 6 ablation studies (including 3 in Appendix D) to demonstrate the robustness of the algorithmic design. The breadth of empirical evaluation is explicitly acknowledged by both Reviewer SfBQ (“Quality” in strength) and Reviewer 16Kv (Strength 2: Good empirical coverage). We also provide our code for reproducibility.

[1] Bickford Smith et al (2025). Rethinking aleatoric and epistemic uncertainty. ICML 2025.

---

### Meta-Review · Area_Chair_Rx3z · 2026-01-07

**Summary:**

This paper proposes ESI, a grey-box uncertainty estimator for LLMs based on instability of token-level predictive distributions under semantic-preserving input interventions. Reviewers agreed that the method is simple, computationally efficient, and empirically well tested, with a clear motivation and broad experimental coverage. However, significant concerns remain regarding the conceptual alignment between the proposed epistemic uncertainty framing and the implemented estimator, the reliance on token-level rather than semantic-level variation, and the degree to which the theoretical justification meaningfully supports the practical score. While the rebuttal clarified scope and added experiments, the core methodological and conceptual issues remain substantial, and relative to other submissions, the paper requires too many interpretive concessions to justify acceptance.

**Reviewer Concerns:**

Several reviewer concerns were partially addressed by the rebuttal and revision. The authors added additional evaluation metrics beyond AUROC (AURC, AUCPR, selective accuracy), incorporated stronger response-level baselines such as Semantic Density, clarified notation separating the KL-motivated theoretical quantity from the practical estimator, and explicitly acknowledged bias and approximation in the derivation. These changes strengthen the empirical case and improve transparency.

However, key concerns remain outstanding. Multiple reviewers continue to question whether token-level distribution shifts are an adequate proxy for semantic uncertainty, especially given that different token sequences can represent the same meaning. The gap between the idealized epistemic quantity discussed in the theory and the practically implemented score (anchoring, single decoded path, Hellinger distance, top-k truncation) remains conceptually large, even if empirically motivated. One reviewer fundamentally disagrees with the epistemic uncertainty framing and argues that the method is not theoretically sound in this respect; while this is partly a philosophical disagreement, it highlights that the paper’s main conceptual claims remain contentious. Overall, the rebuttal improves the paper but does not fully resolve the core concerns around conceptual grounding and interpretability.

**Reviewer Scores:**

Reviewer 1gWk (original score: 6): Likely remains around 6. The added baselines and clarifications address several concrete weaknesses, but the central limitation regarding semantic vs token-level uncertainty remains.

Reviewer 16Kv (original score: 4): Likely increases slightly to around 5. The clarification of approximation bias, added evaluation metrics, and additional ablations address many of the reviewer’s technical questions, though concerns about theoretical alignment persist.

Reviewer SfBQ (original score: 2): Likely remains at 2, or at most increases marginally to 3. While additional metrics address one explicit concern, the reviewer’s core objections to the epistemic framing and semantic interpretation remain largely unchanged.

---

### Decision · Program_Chairs · 2026-01-26

Reject